# Nuclear gene proximity and protein interactions shape transcript covariations in mammalian single cells

Marcel Tarbier [1], Sebastian D. Mackowiak[1,8], João Frade [2,8], Silvina Catuara-Solarz[2], Inna Biryukova [1], Eleni Gelali [3], Diego Bárcena Menéndez[2], Luis Zapata [2,4], Stephan Ossowski[2,5,6], Magda Bienko [3], Caroline J. Gallant[7] & Marc R. Friedländer [1✉]

Single-cell RNA sequencing studies on gene co-expression patterns could yield important regulatory and functional insights, but have so far been limited by the confounding effects of differentiation and cell cycle. We apply a tailored experimental design that eliminates these confounders, and report thousands of intrinsically covarying gene pairs in mouse embryonic stem cells. These covariations form a network with biological properties, outlining known and novel gene interactions. We provide the first evidence that miRNAs naturally induce transcriptome-wide covariations and compare the relative importance of nuclear organization, transcriptional and post-transcriptional regulation in defining covariations. We find that nuclear organization has the greatest impact, and that genes encoding for physically interacting proteins specifically tend to covary, suggesting importance for protein complex formation. Our results lend support to the concept of post-transcriptional RNA operons, but we further present evidence that nuclear proximity of genes may provide substantial functional regulation in mammalian single cells.

[1] Science for Life Laboratory, Department of Molecular Biosciences, The Wenner-Gren Institute, Stockholm University, Stockholm, Sweden. [2] Centre for Genomic Regulation (CRG), The Barcelona Institute for Science and Technology, Barcelona, Spain. [3] Science for Life Laboratory, Department of Medical Biochemistry and Biophysics, Karolinska Institute, Stockholm, Sweden. [4] Center for Evolution and Cancer, The Institute of Cancer Research, London, UK. [5] Department of Experimental and Health Sciences, University Pompeu Fabra, Barcelona, Spain. [6] Institute of Medical Genetics and Applied Genomics, University of Tübingen, Tübingen, Germany. [7] Department of Immunology, Genetics and Pathology, Uppsala University, Uppsala, Sweden. [8] These authors contributed equally: Sebastian D. Mackowiak, João Frade. ✉email: marc.friedlander@scilifelab.se

Two genes that increase or decrease coordinately in expression over multiple conditions are said to covary. Gene expression covariation can be studied over two conditions (e.g., healthy and diseased tissue), in time-series experiments, or in metastudies spanning hundreds of tissues and cell types, for instance from public expression repositories[1–3]. Over the last 20 years, such studies have yielded numerous important biological insights due to the fact that covarying genes are often functionally related, and commonly share the same gene regulatory mechanisms.

In the last ten years, single-cell sequencing methods have emerged, making it possible to profile the entire transcriptomes of individual cells[4–6]. This makes it possible to identify the genes that covary in expression across individual cells, considering in effect every cell as a distinct condition. This research direction holds great promise, since it could reveal biological covariations that are not detectable in analyses of bulk cell populations. First, differences in cellular compositions between samples may disturb covariation analyses in bulk tissues[7]. And second, transcripts can appear to be constantly and moderately expressed in all studied tissues or cell cultures, but may in fact display temporally fluctuating and covarying expression in single cells. This type of covariation may never be detected in bulk tissues. Until now, however, transcriptome-wide single-cell studies of such intrinsic gene covariation patterns have been limited by confounding factors such as cell cycle progression and cell differentiation, which are extrinsic to the genes of interest[8,9]. These confounding factors have a strong impact on the global covariation patterns and could overshadow the more subtle—and potentially more interesting—underlying patterns.

Here, we apply carefully designed experimental conditions to remove the confounding extrinsic effects of differentiation and cell cycle progression, and apply sensitive Smart-Seq2 single-cell sequencing to profile the transcriptomes of hundreds of mouse embryonic stem cells (mESCs). Specifically, using stringent cut-offs we report >67,000 gene pairs that intrinsically covary in expression—more than have been described in previous single-cell studies. These covarying gene pairs interlink to form a network with well-established biological features, following a so-called power-law distribution, and recover known regulatory patterns and pathways. We further apply a novel computational framework to study the relative importance of distinct regulatory mechanisms for gene expression covariation, and find that genes regulated by the same transcription factors or miRNAs tend to covary. We validate that a subset of the covariations is directly induced by miRNAs by repeating our entire experiment in miRNA-deficient cells. The strongest effect, however, is seen between genes that are in nuclear proximity on the same chromosomes, and a similar but weaker effect is seen for genes that are in nuclear proximity but located on distinct chromosomes.

Finally, we test two competing hypotheses regarding the putative function of these gene expression covariations. The first hypothesis states that genes covary in expression to ensure stoichiometric abundances of proteins that function in the same pathway, while the second hypothesis proposes that covariations are important for proper stoichiometry of proteins that are part of the same complexes. We find that covarying genes only tend to share the same function if their encoded proteins also physically interact, lending evidence to the protein complex hypothesis.

In summary, we have combined single-cell RNA sequencing with a tailored experimental design and computational framework to quantify regulatory drivers in single mammalian embryonic stem cells, highlighting the importance of nuclear proximity for gene expression covariations. Additionally, we present evidence that these covariations play a role in ensuring stoichiometry between interacting proteins.

## Results

**Smart-Seq2 sequencing of mouse single-cell transcriptomes**. To obtain reliable and reproducible measurements of gene expression for our study, we applied the Smart-Seq2 protocol to sequence the transcriptomes of 567 individual mouse embryonic stem cells divided between three well-plates which serve as biological replicates (Supplementary Table 1). While labor-intensive and not easily scalable, Smart-seq2 is highly sensitive and precise[6,10,11]. It also reliably detects both exons and introns, which is useful for distinguishing between transcriptional and post-transcriptional regulation[12]. We performed strict quality filtering on the initial set of cells, which resulted in a total of 355 cells considered (see "Methods" section, Supplementary Fig. 1 and Supplementary Table 2). Gene expression values in each cell were normalized to the sum of mRNA sequence reads in the given cell (see "Methods" section, Supplementary Fig. 2), and only genes that displayed substantial biological variation above technical noise, as estimated by artificial ERCC spike-ins (see "Methods" section, Supplementary Fig. 3), were retained. Overall, our analysis yielded reliable gene expression measurements for 8989 genes (Supplementary Table 3 and Supplementary Data 1).

**Homogenous cell population unconfounded by dynamic processes**. For the sequencing experiment, we took several precautions to eliminate the confounding extrinsic effects of cell cycle and differentiation. First, all cells were cultured in 2i+LIF medium, which is a well-established protocol to maintain embryonic stem cells in a homogeneous pluripotent state—excluding potential differentiation effects[13]. Second, we used fluorescence-activated cell sorting to specifically select cells in G2/M phase of the cell cycle, thus excluding major cell cycle effects. This exact combination of growth medium (2i + LIF) and cell cycle phase (G2/M) has been reported to generate particularly homogeneous cell populations with regard to their transcriptome signatures[9]. Indeed, our cell population forms a single cluster when common dimensionality reductions are applied (Supplementary Fig. 4). Using published marker genes, we confirmed that our cells were in the correct cell cycle phase[13,14] and expressed pluripotency but not differentiation marker genes[9,15] (Supplementary Fig. 5). Altogether our cells comprise a homogenous population, unconfounded by cell cycle or differentiation effects.

**Discovery of >67,000 significant positive and negative gene covariations**. To study pairwise gene covariations we calculated Spearman's rank correlation coefficient for all possible gene pairs. We chose this procedure for its ability to detect nonlinear monotonous dependencies and for its robustness towards outliers (see "Methods" section). The measured correlation coefficients were centered around zero (Fig. 1a, left), indicating the absence of overall confounding factors. Importantly, the observed coexpression values had a greater spread than those of permuted controls (Fig. 1a), suggesting the presence of numerous nonrandom biological covariations. Sixty-seven thousand three hundred and twenty-eight gene pairs were considered significantly covarying (42,938 positively and 24,390 negatively) after stringent covariation calling (see "Methods" section). We randomly permuted the count matrix one thousand times and found that the highest number of significant covariations observed was ~2000, which corresponds to only 3.1% of the covariations observed in the original data (see Online Methods, Supplementary Table 4 and Supplementary Data 2). As an additional benchmark, we performed correlation calculations on the pooled replicates and applied multiple-hypothesis testing (Benjamini–Hochberg). Around 90% of the covariations called by our approach are

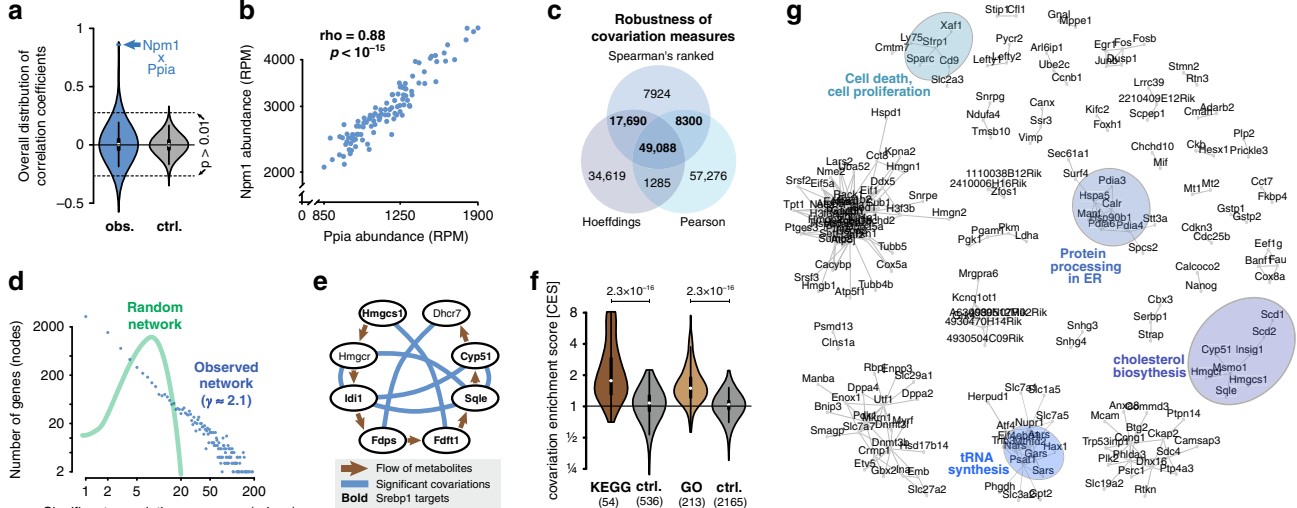

**Fig. 1 Covariation network reflects biological features. a** Transcriptome-wide covariation (co-expression) values for all possible gene pairs. Violin plot of Spearman's ranked statistics (rho-value) for the entire transcriptome (blue) and for a permuted control matrix (gray). Value for the gene pair *Npm1—Ppia* is highlighted. Arrows indicate at which rho-value *p*-values become smaller than 0.01 (rho ~ 0.253). **b** Covariation of the genes *Ppia* and *Npm1*. Abundances for the two genes are in reads per million (RPM) and plotted in log scale. Each data point represents their respective measurement in the same single cell. Spearman's ranked correlation was applied. **c** Spearman's ranked coefficients are in accordance with other covariation and dependency measures. **d** Gene covariation network is scale-free ($\gamma \approx 2.1$). Number of significant covariations per gene against the number of genes with that number of covariations (blue points). Green line illustrates the degree distribution of a random network with same number of genes (nodes) and covariations (edges) as the observed network. **e** Cholesterol biosynthesis pathway is highly enriched for gene pair covariations. Genes involved in cholesterol biosynthesis from acetyl-CoA. Only genes that were robustly detected in our sequencing data are shown. Arrows indicate the flow of metabolites, lines indicated significant covariation between genes. Gene names in bold indicate direct targets of Srebpf1, a transcription factor that is well known to regulate cholesterol biosynthesis. **f** Gene sets that share functional annotations are enriched for covariations. Gene covariation enrichment scores (CES) for gene sets sharing the same gene ontology or sharing the same KEGG pathway annotation as well as respective controls (*p*-values represent a two-sided independent two-group *t*-test). Gene covariation enrichment scores indicate the ratio of observed significant covariations relative to the amount of expected covariations (see main text). **g** Example subnetworks (subset of significant covariations, selected functional subnetworks are highlighted).

supported by pooled and corrected covariations. In fact, our approach is stricter than simple multiple-hypothesis testing since fewer gene pairs are considered significant. An example of a highly significant gene pair is shown in Fig. 1b, wherein each data point represents expression measures in one individual cell. Significant covariations identified using Spearman's ranked correlation coefficient have a high overlap with those retrieved by Pearson's correlation coefficient and with dependency measures recovered through Hoeffding's D statistics (Fig. 1c), showing the robustness of the approach. Finally, we validated several of the gene expression covariations using single-molecule FISH (Supplementary Figs. 6, 7) and single-cell quantitative RT-PCR (Supplementary Fig. 8). In summary, we present >67,000 high-confidence gene pair covariations—more than have been reported in previous single-cell studies.

**Covariation network features reflect biological functions**. We observed that the covarying gene pairs link together in complex patterns that can be described as a network. It is well-established that biological networks, such as those arising from transcription factor targeting or protein interactions, have properties that differ from those of random networks[16]. For instance, biological networks tend to be scale-free following a so-called power law distributions, such that most genes only have few interactions with other genes while few genes represent hubs in the network, interacting with many other genes. Consistent with our network having biological rather than technical origins, we found that our covariation network follows such a power law distribution ($\gamma \approx$ 2.1, Fig. 1d, blue). Importantly, this network structure is distinctly different from that of a random network with the same overall

connectivity (Fig. 1d, green). Further network features are listed in Supplementary Fig. 9c.

Within this covariation network we identified many biologically meaningful subnetworks, such as the one formed by genes involved in cholesterol biosynthesis (Fig. 1e). These genes are known to be activated when the SREBF1 transcription factor is cleaved from the Golgi membranes, and shuttled to the nucleus in response to lack of cholesterol[17] and can therefore be expected to covary in expression, depending on the localization of SREBF1 protein. Another notable subnetwork is formed by genes involved in the formation of the TCP1 ring complex, a chaperone involved e.g. in tubulin biogenesis[18] (Supplementary Fig. 10).

A substantial proportion of the observed covariations (~14,500 gene pairs, not included in overall counts listed above) are between ribosomal proteins. These covariations have previously been reported for bulk cell populations[19] and likely have functions in proteostasis[20]. It was recently reported that four of these proteins (RPL10, RPL38, RPS7, and RPS25) are optional components of the ribosome whose inclusion or exclusion can influence which pools of transcripts are preferentially translated[21]. We find that these four ribosomal proteins all covary positively and significantly with each other, providing evidence that they may not function by a mutually exclusive either-or logic in single mouse embryonic stem cells in steady state condition. In the following sections, we have excluded ribosomal protein genes and focus on other types of covariations.

Applying our method for measuring covariation enrichment over large gene sets (see section of CES score below), we find that genes sharing common Gene Ontology terms are 1.47-fold more likely to be covarying (we observe 47% more significant covariation than we would expect by chance), while permuted control sets show no such

enrichment (Fig. 1f). The same holds true for genes sharing common a KEGG pathway annotation, where the enrichment is 1.86-fold (Fig. 1f). We reason that genes sharing functions or pathways are more likely to be regulated in a similar fashion and thus tend to covary. In conclusion, the covarying gene pairs form a comprehensive scale-free network which is associated with annotated cellular functions and pathways.

**Covariations retrace known aspects of stem cell biology.** The pluripotency of mouse embryonic stem cells has been studied extensively and several studies focus on characterizing their transcriptomes and gene regulatory circuits[13,19,22–24]. The network we observe recapitulates many known relationships between pluripotency markers in mouse embryonic stem cells. For instance, positive covariations support the activation of *Fgf4* through *Nanog* and *Sox2*[25,26], while negative covariations support the inhibition of *Dnmt3a/b/l* by *Prdm14*[15,27] and of *Dppa3* by *Tbx3*[28]. While our data support previous claims that *Nanog* is positively covarying with *Klf4, Sox2, Tet2,* and *Kat6b*[9], we see little support for a covariation with *Esrrb, Zfp42* and *Tet1*, and we observe significant negative covariations with *Pou5f1* and *Dnmt3a* in single cells. With regard to predicted pluripotency genes, we can confirm that there are strong covariations between *Etv5* and weak covariations between *Ptma,* and *Zfp710* and other pluripotency genes. Covariations of pluripotency genes can be found in Supplementary Table 5. In summary, the detected covariations are in accordance with known gene expression patterns in stem cell biology and give hints at new connections.

**Covariation enrichment score (CES).** To systematically investigate the functional and regulatory implications of expression covariations, we defined the CES for gene sets of interest. It indicates whether for a given gene set, we observe fewer or more significant covariations between the genes than we would expect based on a simple background model. The CES provides an easily interpretable single metric—fold-enrichment rather than coefficients and *p*-values, which can be difficult to interpret. It also allows for easy visualization and comparison of covariations in gene sets.

Our background model considers the total number of significant covariations for each gene as well as the number of covariations of all its potential pairing genes. In other words, it is the factor of the probabilities of two genes if their covariations were distributed randomly [Eq. (1)], summing over all possible pairs in the gene set (Supplementary Fig. 11).

$$P(\text{sigCov}(g_a, g_b)) = \frac{\sum_{i=1}^{N} \text{sigCov}(g_a, g_i) \times \sum_{j=1}^{N} \text{sigCov}(g_b, g_j)}{\sum_{i=1}^{N} \sum_{j=1}^{N} \text{sigCov}(g_i, g_j)}$$

$$(1)$$

We can then test whether genes that are regulated by the same regulatory factor, e.g., a transcription factor or a miRNA, tend to covary as a consequence of varying abundance or activity of said factor in individual cells.

**MiRNAs induce transcriptome-wide gene expression covariation.** We first apply the CES to study the regulatory impact of miRNAs, which are important post-transcriptional regulators of gene expression[29]. In most conditions, these small RNAs downregulate the expression of protein coding genes by binding their mRNA transcripts and leading to their degradation[30]. This targeting takes place in the cytoplasm and is therefore spatially decoupled from transcriptional regulation.

We speculate that miRNA regulation of gene expression may be a source of gene covariations. For instance, if a miRNA is highly abundant in a given cell, its targets may be coordinately

repressed, and we expect an enrichment of covariations across single cells for these targets. To test this hypothesis, we investigated the top-ranking miRNA targets according to TargetScan[31], the most widely used catalog of miRNA-target interactions. In this study, we focused on the seven most highly expressed conserved miRNA families (including the miR-15 and miR-290 families) in mouse embryonic stem cells.

Strikingly, miRNA gene target sets are significantly enriched for gene covariations. In median the top 200 targets of each of the seven miRNA families are 28% more likely to covary with each other than expected ($p = 0.032$). The enrichments exhibit a gradient such that the top-ranking targets show a stronger enrichment in comparison to sets that include lower ranking targets (Fig. 2a). As introns are spliced out in the nucleus, their abundances cannot be impacted by miRNA action in the cytosol. Consistent with this, miRNA targets do not significantly covary at the intron level (Fig. 2b).

To exclude the possibility that these covariations originate from other post-transcriptional effectors, we investigated cells that are void of canonical miRNAs. DROSHA is an endonuclease involved in the biogenesis of miRNAs without which canonical miRNAs cannot be produced. We used an inducible *Drosha* knock-out cell line to validate the miRNA dependence of these covariations (see "Methods" section) and sequenced the transcriptomes of 343 of these knock-out cells using clonal expansion from a single cell and sorting of cells in G2/M phase as described above. We have previously demonstrated the global loss of miRNAs in this particular cell line[32]. Furthermore, predicted miRNA targets are specifically upregulated in cells void of miRNA as a result of their de-repression (Supplementary Fig. 12). As expected, there is no covariation enrichment in miRNA target sets in *Drosha* knock-out cells (Fig. 2b), demonstrating that these covariations are directly caused by miRNA activity.

We additionally investigated what we call the reverse covariation enrichment. Here, we observe whether significantly covarying gene pairs are regulated by the same miRNA more often than a permuted background set (see "Methods" section). We find that covarying genes are 12% more likely to be co-regulated by the top 16 miRNAs and 35% more likely to be regulated by the seven most highly expressed conserved miRNAs (Fig. 2c), showing the importance of miRNA conservation and abundance in inducing covariations. It has previously been reported that individual miRNAs can induce gene covariations[33], but here we show that this in fact holds true for many miRNAs, transcriptome-wide. We also present evidence that natural (noninduced) fluctuations of miRNA abundance or activity are sufficient to cause gene expression covariations.

From a network perspective, we found that >6000 high-confidence gene covariations were lost in the cells devoid of miRNAs, while less than 3000 new covariations were gained (Supplementary Fig. 9a). A substantial number of the genes that ceased to covary were miRNA targets and the ratio of lost to gained covariations increased when high-confidence targets were considered (Supplementary Fig. 9b). The genes that lost covariations were enriched in functions in RNA biology (Supplementary Fig. 9d), including *PolII* regulation. The average number of covariations per gene decreased significantly in the miRNA-depleted cells, from 10.1 to 8.4 covariations, and the number of genes without covariations increased from 2265 to 2866 (Supplementary Fig. 9c). Overall, this indicates a global loss of gene expression coordination in cells devoid of miRNAs.

**Genes regulated by the same TFs covary with each other.** To investigate how regulation by transcription factors influences covariation patterns, we studied the binding sites of 145

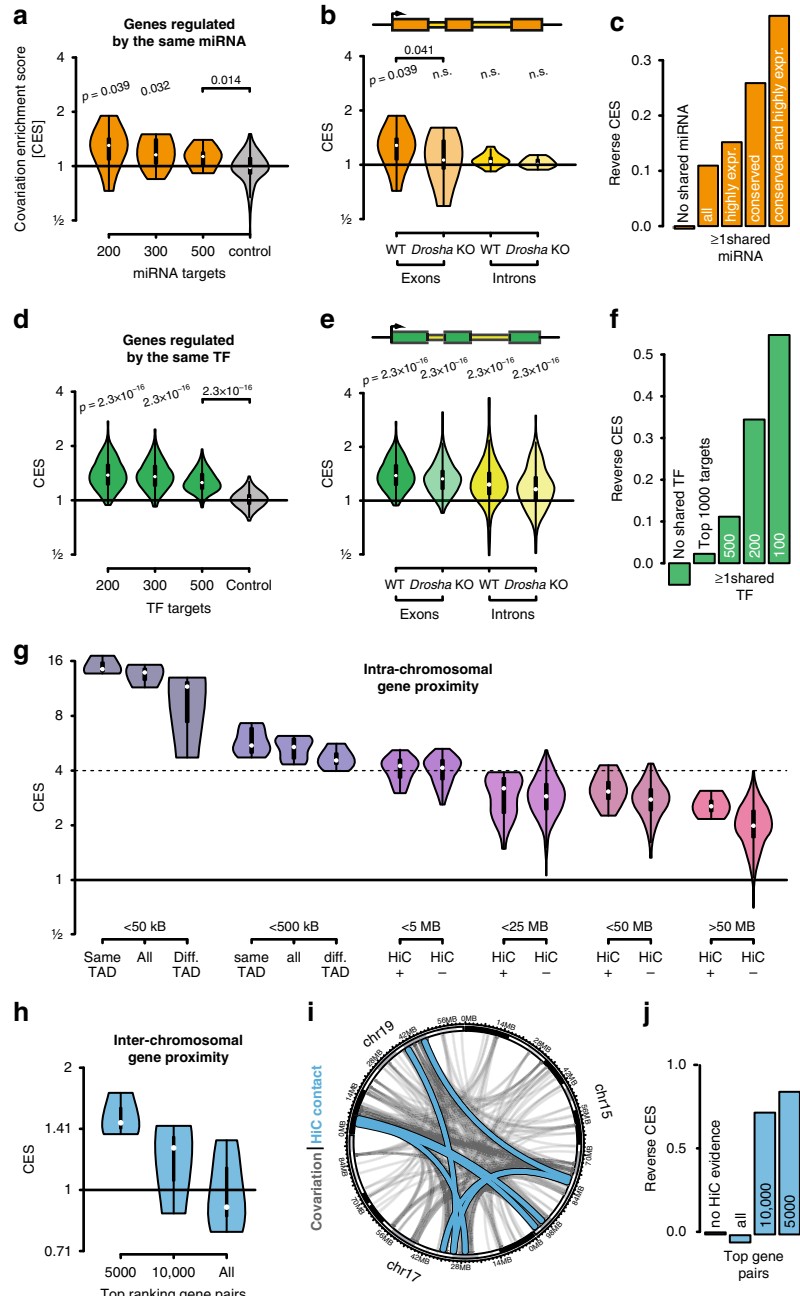

**Fig. 2 miRNAs, transcription factors and nuclear organization define covariations. a** miRNA targets tend to covary. Covariations enrichment scores (CES) for the top 200, 300, and 500 ranked miRNA targets according to TargetScan, for the seven highest expressed conserved miRNA and for a control set for comparison for 500 randomly selected targets. *p*-values refer to respective controls. **b** miRNA target covariations occur post-transcriptionally and are miRNA-dependent. Enrichment in sets of the top 200 ranked miRNA targets in parental cells (WT) and *Drosha* KO cells, that are void of canonical miRNAs. Enrichments are color coded for exonic reads, representing post-transcriptional regulation (orange) or intronic reads representing transcriptional regulation (yellow). **c** Covarying genes are enriched for shared miRNA targeting. Reverse covariation enrichment shows the log2 ratio between covariations that share a common miRNA and permuted covariations that share a common miRNA. **d** Transcription factor targets are enriched for gene covariations. Enrichment in sets of the top 200, 300, and 500 transcription factor targets, for 145 transcription factors profiled with ChIP-seq. Control for comparison is shown for 500 randomly selected targets. *p*-values refer to respective controls. **e** Transcription factor target covariations are transcriptional and miRNA-independent. Enrichment in sets of the 200 ranked transcription factor targets in parental cells (WT) and *Drosha* KO cells. Enrichments are color coded for exonic reads (dark green) or intronic reads (light green). **f** Covarying genes are enriched for shared transcription factor targeting (figure similar to c). **g** Genes that are in close nuclear proximity and locate to the same chromosome are enriched for covariations. The range categories are mutually exclusive, for instance pairs of genes that are <5 MB apart are not included in the <25 MB category. **h** Gene regions that are in close nuclear proximity and locate to different chromosomes are enriched for covariations. Since relatively few intrachromosomal Hi-C contacts were identified, we here used a less stringent criteria (*p*-value <0.05) cut-off to robustly identify significant covariations. **i** Circos plot showing significant covariations and Hi-C contacts for chromosomes 15, 17, and 19. Significantly covarying gene pairs are connected by a light blue line. Inter-chromosomal Hi-C contacts are shown as gray lines. **j** Covarying genes are enriched for interchromosomal Hi-C contacts (figure similar to c). **a**, **b**, **d**, **e** All *p*-values represent two-sided independent two-group *t*-tests between targets sets and respective controls (see "Methods" section).

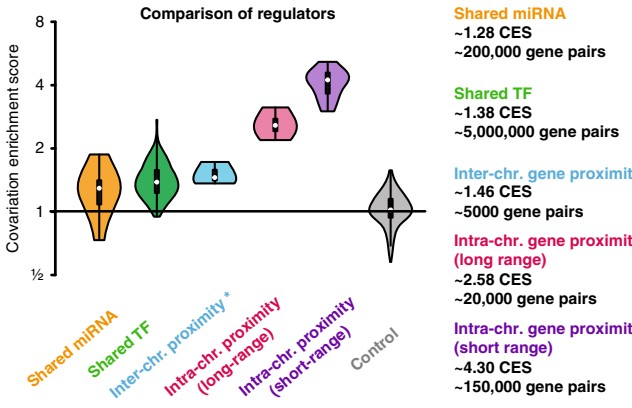

**Shared miRNA**
~1.28 CES
~200,000 gene pairs

**Shared TF**
~1.38 CES
~5,000,000 gene pairs

**Inter-chr. gene proximity**
~1.46 CES
~5000 gene pairs

**Intra-chr. gene proximity (long range)**
~2.58 CES
~20,000 gene pairs

**Intra-chr. gene proximity (short range)**
~4.30 CES
~150,000 gene pairs

**Fig. 3 Relative importance of miRNAs, transcription factors and nuclear proximity for covariations.** Comparison of covariation enrichment (CES) scores for genes that are either regulated by the same miRNA, regulated by the same transcription factor or that are in nuclear proximity—divided into intrachromosomal and interchromosomal pairs.

transcription factors for which mouse ES cell ChIP-seq data were deposited in the Cistrome database[34]. As for miRNAs, we observe a gradient in covariation enrichment which is stronger for the top-ranking transcription factor targets compared to lower ranking targets (Fig. 2d). Importantly, transcription factor target sets are significantly enriched for gene covariations both on the exon and the intron level (Fig. 2e), consistent with transcriptional regulation. In median, the top 200 ranked targets of these transcription factors 1.38-fold enriched for coexpression on the exon level ($p$-value $<2.3 \times 10^{-16}$) and 1.22-fold enriched for coexpression at the intron level ($p$-value $<2.3 \times 10^{-16}$). As expected, the covariation enrichment of transcription factor targets is not significantly lowered in *Drosha* knock-out cells (Fig. 2e). In conclusion, genes that are regulated by the same transcription factor tend to covary, possibly due to stochastic variations in transcription factor abundance and activity between individual cells. This effect acts on millions of gene pairs while the mean magnitude of the regulation is similar to what we describe for miRNA-specific regulation.

**Proximal genes on the same or different chromosomes covary.** Genes that neighbor on the same chromosome are known to show coexpression[35]; this also holds for genes within the same chromatin loop or within the same topologically associated domain (TAD). Furthermore, the concept of transcription factories covers dynamically assembled complexes that facilitate transcription, and are dependent on intrachromosomal or interchromosomal interactions[36]. To investigate the covariation enrichment on genomic regions that are in proximity within the nucleus, we analyzed mouse embryonic stem cell Hi-C-seq data[37,38]. From here on, we define proximal genes as those whose interaction is supported by Hi-C data, whether the interaction is intrachromosomal or interchromosomal (see "Methods" section). Our data shows that genes which are proximal and located on the same chromosome are highly enriched for covariations (Fig. 2g). Genes that are close in linear distance on the chromosome (<5 MB) are enriched ~4-fold in covariations, while genes that are distal (>50 MB) are enriched 2.1-fold. This observation is robust to changes in the computational analysis and normalization (Supplementary Figs. 13, 14). The effect is also detectable at the intron level, confirming an origin in transcriptional regulation at the level of nascent transcripts. Genes that are on the same chromosome are almost twice (1.9-fold) as likely to covary than

expected, even when their proximity is not supported by Hi-C (Fig. 2g, far right). The highest enrichment was detected for genes that are both close in linear distance on the same chromosome and are predicted to be in the same TAD, which are ~15-fold more likely to covary (Fig. 2g, far left). Intriguingly, proximal genes on different chromosomes also show substantial covariation enrichment (Fig. 2h–j), supporting the notion of transcription factories that incorporate areas from multiple chromosomes.

Next, we ranked the relative importance of transcription factors, miRNAs, and nuclear proximity for the regulation of covariation (Fig. 3). We observed that miRNA targets were 1.28-fold, transcription factor targets were 1.38-fold, genes in nuclear proximity on different chromosome were 1.46-fold, and, remarkably, genes that are proximal on the same chromosome are 4.3-fold more likely to covary. While the exact enrichments are likely specific to the study system and target selection strategy, it is clear that transcriptional regulation, miRNA-mediated regulation and, surprisingly, interchromosomal nuclear proximity all play important roles. In our setting, however, intrachromosomal nuclear proximity is the strongest predictor of gene expression covariations.

**Protein interaction drives gene covariations.** Next, we examined putative functions of the covariations that we observe in single cells. We formulated two hypotheses. The first hypothesis is the pathway hypothesis: that genes involved in the same pathway are coordinated in expression, for instance to avoid bottlenecks in the production of metabolic intermediates[39]. The second hypothesis is the complex hypothesis: that covariations ensure correct stoichiometry among proteins that are part of the same heteromeric protein complex, since surplus proteins may misfold or even cause aggregates[40].

As previously stated, genes that share the same Gene Ontology function or process or the same KEGG pathway annotation are significantly enriched for gene covariations (Fig. 1f). The same is true for genes that physically interact on the protein level according to experimental evidence gathered by the STRING database (Fig. 4a). For these interactions we observe a gradient in which those interactions with the highest confidence/affinity score also have the highest enrichment for covariations, consistent with previous findings in single cancer cells[41]. We then determined that genes that contribute to the same complex are 2.6-fold more likely to covary, compared to just 1.6-fold for genes that are part of the same pathway (Fig. 4b), lending support to the complex hypothesis.

To further test the two hypotheses, we split the set of functionally related genes into one set with genes that share a functional annotation as well as protein interaction and those that share a functional annotation but no protein interaction. If the pathway hypothesis holds, we would expect both gene sets to covary, since they share functions. If the complex hypothesis holds, we would expect only the genes whose proteins physically interact to covary, since the covariations are needed for proper stoichiometry of proteins in the complexes. Strikingly, when genes that physically interact at the protein level are excluded from the analysis, we find no covariation enrichment for either of the GO and KEGG functional annotations (Fig. 4c). In other words, there is no indication of covariation enrichment for proteins in the same pathway without evidence that they physically interact. Altogether, our data suggests that direct interactions between proteins in the same complex, rather than pathway stoichiometry, (Fig. 4b–d) as a selector for covariations.

**Predictive power of gene covariations.** We next investigated if our observed covariations can be used to predict genes that share

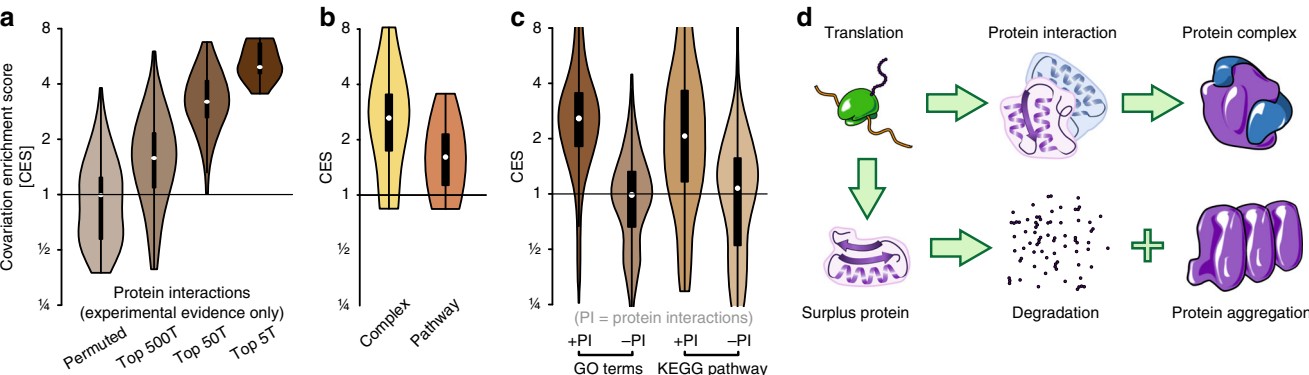

**Fig. 4 Proteins that physically interact are specifically enriched for covariations at the RNA level. a** Genes that interact on the protein level are highly enriched for covariations. Covariation enrichment of genes that are annotated to be interacting on the protein level according to the STRING database. **b** Covariations enrichment for genes whose protein products are part of the same physical complex and for genes that are part of the same reaction (pathway). **c** Gene covariations are mainly driven by protein interaction. Covariation enrichment of genes sharing the same GO annotation or KEGG pathway annotation. GO and KEGG annotations are stratified into pairs with shared annotation and experimentally identified protein interaction (+PI) and shared annotation but lack of experimentally identified protein interaction (−PI). **d** Model. Heteromeric protein complexes require proper stoichiometry of protein components. Proteins that are in surplus can be degraded, misfolded or form aggregates.

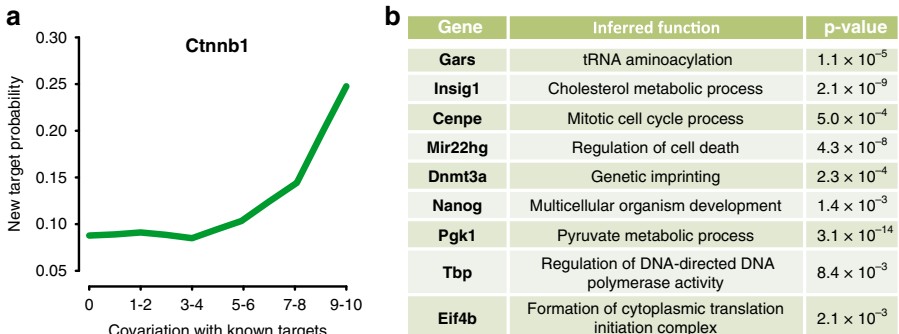

**Fig. 5 Gene covariation information can predict regulatory targets. a** Genes that covary with transcription factor targets are likely targets of the same factor (Ctnnb1) and can be validated by ChIP-seq. Probability for genes that share a certain number of significant covariations with the top 100 targets identified via ChIP-seq to be identified de novo in an independent second ChIP-seq experiment. **b** Examples of genes whose canonical function was retrieved purely from the functional annotations of covarying genes (*p*-values determined by Fisher's exact test).

upstream regulators through a guilt-by-association principle. We hypothesize that if a gene of interest covaries with numerous known targets of a transcription factor, it is likely a target of said factor. To test this hypothesis, we noted all genes that had been identified as a targets of the transcription factor *Ctnnb1* in a mouse ES cell ChIP-seq experiment[42]. This gene is known to regulate cell adhesion and has been linked to various cancer phenotypes[43]. We then ranked all other genes according to how many of the top 100 *Ctnnb1* targets they covary with, and observed that the more covariations a gene exhibited, the more likely it is to be bound by *Ctnnb1* in a second ChIP-seq experiment[42] (Fig. 5a). In other words, the more significant covariations a gene had with the high confidence targets identified in the first experiment, the more often it was observed among the high confidence targets identified in a second independent experiment. While the predictive power of our method is limited (target probability ~25%, Fig. 5a) it serves as a proof of principle that single-cell transcriptome data can be used for predicting regulatory relations even in a homogeneous cell population. This approach could be used to make sparse data sets more complete, through guilt-by-association with previously identified targets or to identify targets that escape current technologies due to biases. Last, we found that the function of genes could be inferred by surveying functional annotations of covarying genes (Fig. 5b, see "Methods" section). This may not only aid functional annotation

but could also reveal hidden gene functions, so-called moonlighting. In conclusion, knowledge of gene covariations across homogeneous single cells can be used to infer gene function and regulation through associations.

## Discussion

We show that statistically robust and biologically meaningful gene covariations that can be detected in homogeneous non-dynamic single cell populations. Evidence to support this claim include the validation by statistical methods, a low estimated false discovery rate, the recovery of known regulatory patterns, and a power-law distribution of network edges commonly found in biological networks. Our experimental set-up allows for the study of widespread gene expression covariations unrelated to cell cycle and other dynamic changes in the cells such as differentiation. Strikingly, all major regulatory mechanisms—post-transcriptional, transcriptional and by nuclear proximity—influence covariation patterns. We experimentally confirmed the importance of post-transcriptional regulation through miRNAs by showing that depletion of miRNAs results in a specific loss of a subset of covariations.

Based on our findings, we propose a hierarchy of gene covariation regulation in mESCs. We place regulation via intrachromosomal proximity first due to the strength of the effect, and

transcription factors second because of the size of the affected target pool (Fig. 3). The influence of interchromosomal proximity and miRNA regulation is comparatively smaller though still substantial.

As targets of the same regulator tend to covary as well as genes that are part of the same functional units, covariations can be used to predict gene function and regulation. We show that we not only recover known gene functions and transcription factor targeting but, as a proof of principle, also demonstrate the predictive potential for both gene function and regulation.

Importantly, we find that covarying genes only tend to share the same function if their encoded proteins also physically interact, suggesting a role in protein complex stoichiometry. The induction of gene expression covariation could be beneficial to cells as it is well understood that the formation of heteromeric protein complexes is often needed for proper folding and for the stability of the proteins involved[44]. In bacteria, spatial separation of the translation of such proteins leads to misfolding events[45]. It is conceivable that temporal separation might result in similar effects. The production of misfolded proteins that must be removed by degradation is costly from an energetic point of view, and the accumulation of misfolded protein can have lethal consequences for cells (Fig. 4d). We suggest, therefore, that establishing expression covariation of such genes already on the RNA level might be an advantage in evolutionary terms.

In this study, we measure RNA rather than protein with the latter being closer to the cellular phenotype. However, when inferring upstream regulation, it may be more informative to measure RNA. Furthermore, many of the most interesting and biologically meaningful covariations that we discover may not be detectable at the protein level, even in single cells. For instance, transcript covariations may be important for cofolding, but they may not be visible at the proteome level for proteins that have long half-lives and that are therefore more stably expressed. It will be exciting to study covariations at the protein level, when technologies to accurately profile hundreds of proteins in single cells become available.

A caveat of our study is that different published data sources and methods were used to identify miRNA targets, transcription factors targets and genes that are in close proximity, complicating direct comparisons among them. For example, public ChIP-seq was used to infer transcription factor targets, while Hi-C was used to detect genes in the same nuclear vicinity. These methods have distinct limitations and ranges of sensitivity. However, all of the methods we employ are state-of-the-art in their respective fields, and in some cases the methods have limitations even within those areas. For instance, there is evidence that Hi-C may underestimate interchromosomal contacts[46] and it is well-established that even the best miRNA target prediction has imperfect accuracy[47,48], meaning the effects of nuclear proximity and miRNA repression on gene expression covariations may be more profound than we estimate here.

Gene coexpression studies have been conducted on pools of cells for decades, yielding important insights into covariations and network properties. These studies, however, have been limited in their capacity to study changes in network properties following a genetic perturbation. For instance, to study the effects of *Drosha* knockout using pooled cells, it would be necessary to ablate the gene in dozens or hundreds of cell lines in parallel to have the statistical strength to call covariations. In contrast, our study serves as a proof-of-concept that it is possible to delete a gene in a single cell line, and then consider each of hundreds of individual cells as an independent condition, thus obtaining the statistical power to resolve network properties in a single experiment. In our study, we find that many more covariations are lost than gained in the *Drosha* knockout cells, and we observe

a general loss of network connectivity. This highlights the importance of miRNAs in maintaining gene expression synchronicity and global gene network connectivity—an insight that would be difficult to obtain with bulk cell or classical single-gene approaches. In summary, we demonstrate that the combination of single-cell sequencing, gene covariation analysis and genetic perturbations can yield insights into the robustness of regulatory networks with unprecedented ease and depth.

A previous study of RNA and protein covariations using samples from bulk cell populations[35] found that neighboring genes on the same chromosomes are often coexpressed at the RNA level, but are not functionally related and that the covariations do not translate to the protein level. On the contrary, we observe that gene pairs in nuclear proximity that share an interaction on the protein level are in fact 7.5-fold enriched for covariations, suggesting a specific co-occurrence of nuclear proximity, RNA co-expression and shared function. Using a database for bulk cell protein expression covariations[49], we further find that 21% of our observed proximity-related RNA covariations translate to the protein level, compared to 6% for background gene pairs. The apparent contrast between these results may derive from the fact that the previous study was conducted in immortalized primary cell lines from human individuals[35], where genetic variants that strongly impact protein levels may have been specifically selected against by evolution. In contrast, temporal fluctuations of protein levels may be tolerated in individual cells from cell lines, allowing more refined measurements. It is possible that nuclear proximity limits independent regulation of physically close genes rather than enabling active coregulation. Genes that need tight coregulation may therefore evolutionary tend to locate in proximity on the same chromosome. Regardless of the causality, genes that share a protein interaction and therefore need stoichiometric coexpression may be placed in nuclear proximity through evolution. Surprisingly, a recent study provides evidence that chromosome rearrangements do not substantially impact gene expression in *Drosophila*[50]. These findings are not inconsistent with ours, however, and may reflect differences between mammals and invertebrates, or between measuring averages of gene expression in tissues and expression covariations in single cells. Overall, our findings highlight the advantages of studying variation of gene regulation at the single-cell level.

It has been proposed that while prokaryotes use cotranscribed operons to ensure synchronized expression and stoichiometry of proteins in common pathways or complexes, eukaryotes use post-transcriptional regulation to ensure a similar outcome at the RNA level. The integrated effect of dispersed transcription and coordinated post-transcriptional regulation has been named RNA operons or Regulons[51]. Our results support the idea that eukaryotic post-transcriptional regulators such as miRNAs can coordinate gene expression at the RNA level. Finally, we provide evidence that substantial functional regulation occurs at the level of nuclear organization, by genes on the same chromosome or by genes that are in proximity although on distinct chromosomes.

## Methods
**Drosha knock-out**. The *Drosha^F* E14 *129Sv*-derived mouse embryonic stem cell line (mESC) was provided by M. Chong[52]. *Drosha^KO* cells were generated using the *Drosha^F* cell line containing the tamoxifen-inducible LoxP—exon9—LoxP and a neomycin selection cassette. The tamoxifen induction was performed in standard serum-containing media. After 48 h of incubation with tamoxifen, single cells were FACS-sorted, clonally expanded and selected for deletion of exon 9 yielding the null allele of *Drosha*.

**Cell culture**. The mESCs were maintained in (1) standard serum-containing media (DMEM media, Gibco): 1× nonessential amino acids (Gibco), 1000 U ml⁻¹ ESGRO mouse LIF medium supplement (Millipore), 15% heat-inactivated fetal

bovine serum (Gibco), 2 mM glutamine (Gibco), 1 mM sodium pyruvate (Gibco), 0.1 mM b-mercaptoethanol (Gibco), 1× penicillin-streptomycin (Gibco). The standard serum-containing media was supplemented with 250 µg ml⁻¹ neomycin (Sigma) for maintenance of $Drosha^F$ cells; (2) 2i media containing Ndiff227 medium (Cellartis Takara Bio), 1000 U ml⁻¹ ESGRO mouse LIF medium supplement (Millipore), 1 µM PD0325901 (Selleckchem), 3 µM CHIR-99021 (Selleckchem) and 1× penicillin-streptomycin (Gibco) onto feeder-free 0.1% EmbryoMax gelatine (Merck Millipore) coated flasks at 5% $CO_2$ and 37 °C. Cells were tested for mycoplasma contamination and propagated in serum-containing media for three passages before switching to 2i media. After adapting to the serum-free media, the cells were propagated for at least three passages in 2i media and used for single-cell sequencing when 50–70% confluency was reached. To harvest cells, they were incubated with Accutase (Sigma) at 37 °C for 5 min followed by centrifugation.

**Cell sorting**. Cells were resuspended in 2i media (∼10⁶ cells ml⁻¹), DNA stained with 10 µg ml⁻¹ Hoechst-33342 (Sigma) at 37 °C for 15 min, and then stained with 1 µg ml⁻¹ propidium iodide (Sigma) to reveal cell viability. Single cells were sorted in G2/M using a BD Influx (BD Bioscience) into 384-well plates containing 2.3 µl of lysis buffer containing ERCC spiking (1:40,000 dilution) in each well.

**cDNA library preparation and RNA sequencing**. Dual-indexed cDNA libraries were prepared using the Illumina Nextera XT dual index library prep kit following the Smart-Seq2 protocol[6]. cDNA libraries were pooled and 50-bp single-end sequencing was performed on an Illumina HiSeq 2000 platform.

**Read mapping and counting**. An extended mouse genome assembly was created by concatenating the ERCC spike-in sequences (obtained from thermofisher.com) to the mouse genome assembly mm10 (obtained from genome.ucsc.edu). This assembly was indexed using bowtie2 (2.2.3) and reads were mapped to the indexed genome using TopHat (2.0.12). *tophat2–GTF transcriptome.gtf mm10_geno-me_index_dir raw_reads.fastq*. After mapping, PCR duplicates were removed with SAMtools *rmdup* (version 0.1.19–44428 cd). *samtools rmdup -s infile.bam rmdup_file.bam*. For the read count assignment to gene loci, a custom Perl script (available upon request) was used to intersect reads with gene annotation (RefSeq, obtained from genome.ucsc.edu, 28.08.2015). Specifically, to ensure stringency in the covariation analysis, only reads having a unique genome mapping position and gene annotation were considered. Transcript isoforms from the same gene locus were fused so that two overlapping exons from different isoforms were combined into a single exon, which comprises the start of the upstream exon and the end of the downstream exon. Gene models called Gm*Number* (e.g., Gm10024) were excluded. Similar analysis was performed separately for reads mapping to intron annotations.

**Quality control of sequencing data**. Cells were kept for further analysis if they fulfilled the following criteria: cells had more than 200,000 sequenced reads, more than 80% of the reads mapped to the transcriptome or genome, more than 40% of 19,127 genes were detected, the spike-in fraction of reads was less than 1%, the mitochondrial mapped reads were below 5% and the PCR duplicates per cell were below 30% of all reads. This procedure also removed empty wells, duplets and incompletely lysed cells due to their different mapping statistics. Mapping statistics are shown in Supplementary Fig. 1. Additionally, cells that were flagged as S or G1 phase using the *cyclone* function of the SCRAN package for R were excluded[53]. Remaining cells showed homogeneous expression of pluripotency and G2/M-phase markers, while differentiation markers and G1/S-phase markers were largely absent (Supplementary Fig. 5). Individual genes were excluded from analysis if they were not expressed in at least half of all cells in all subsets of conditions (control and Drosha knock-out, respectively) and replicates. This resulted in 8989 genes (9105 in Drosha KO, 8501 in both) that were considered reliably expressed. These genes generally exhibited mean abundances higher than 16 RPM (see below), which we considered to be the cut-off for reliably detected genes above technical background (Supplementary Fig. 3).

**Gene expression normalization**. To ensure that the normalization method did not bias the covariation analyses, the following well-established normalization methods were all tested and considered: normalization by overall read count, overall spike-in count, overall count of endogenous reads, and fraction of spike-ins reads. To avoid over-fitting we further investigated whether the normalization factors which were applied to all observations (individual cells) itself correlated with the features (genes) after the normalization. Normalization methods that induced correlations in our data or failed to remove correlation with technical factors were excluded. Simple normalization by sequencing depth was most suitable for removal of technical effects while avoiding overfitting (Supplementary Fig. 2). For this normalization read counts for each gene were divided by the total sum of mapped reads in the respective cell and multiplied by 10⁶, resulting in reads per million (RPM). No length normalization was applied for two reasons. First, with regard to the ERCC spike-ins an RPKM normalization was found to over-estimate the abundance of short transcripts. Second, when analyzing pair-wise

correlations between genes with a ranked approach the relative abundance of each gene across cells is sufficient.

**Covariation**. Covariations were calculated using Spearman's rank correlation coefficient. p-values were determined via a z-score for the Fisher transformation of the correlation coefficient rho ($\rho$). For a more stringent analysis we only considered genes that were significantly correlated ($p < 0.01$) in at least two of our three replicates. Furthermore, covariations were discarded if the sign of the correlation coefficient differed in one of replicates.

We note that ribosomal proteins make up a sizeable proportion of the positive covariations with 68% of all possible riboprotein gene-pairs being significantly covarying, whereas only 0.3% of all other gene pairs covary. Riboproteins are known to be coordinately expressed, however the underlying regulatory mechanism is not well understood—especially in mammals[54]. We therefore excluded riboproteins from the following analysis, resulting in 67,328 remaining covariations of which 42,938 are positive and 24,390 are negative.

**Covariation enrichment score**. We developed a simple covariation enrichment approach. The probability of a gene pair to be covarying by chance can be calculated by the product sums of significant covariations of each individual gene, divided by the total sum of significant covariations within the whole data set (2). For a set of multiple genes, the sum of individual probabilities will be referred to as "expected covariations".

$$P(\text{sigCov}(g_a, g_b)) = \frac{\sum_{i=1}^{N} \text{sigCov}(g_a, g_i) \times \sum_{j=1}^{N} \text{sigCov}(g_b, g_j)}{\sum_{i=1}^{N} \sum_{j=1}^{N} \text{sigCov}(g_i, g_j)}. \quad (2)$$

This estimation works remarkably well for random gene sets (Supplementary Fig. 11). An enrichment for a gene set of interest can be calculated as the fold-change (or log2 fold-change) of the expected and the observed significant covariations within the aforementioned gene set. As an intuitive control for the CES a reverse enrichment approach is applied. While the CES describes enrichment for covariations in a gene pair set, e.g., genes that are targeted by the same miRNA, the reverse enrichment describes how often a certain gene pair feature, e.g., shared miRNA targeting, is found in the set of all significantly covarying gene pairs in comparison to a permuted set of these pairs.

**Binning approaches**. Covariation enrichment scores can only be calculated on reasonably sized sets of gene-pairs. Each set needs to contain one—preferably multiple—significant covariations. Therefore individual pairs have to be binned together. Depending on the overall sparseness of significant covariations in a certain array of gene pairs varying bin sizes may be applied. Binning is performed in a randomized manner. The details of each binning operation is described in the respective sections.

**Transcription (co-)factor targets**. Transcription factor and cofactor targets were determined by public chromatin immunoprecipitation with massively parallel DNA sequencing (ChIP-seq) data. Transcription factor lists were retrieved from the Cistrome database[55] setting "Mus musculus" for species and "Embryonic Stem Cell" for biological sources. Only sets passing 4 out of 6 listed quality scores were considered. Putative target lists were downloaded and used for analysis. Target rank was determined by the assigned score. Covariation enrichment was calculated for each target set using the indicated number of top-ranking targets. For transcription factors listing multiple putative targets sets from the same or multiple studies enrichments were calculated individually and the median of all enrichments was used as representative for said transcription factor.

**MiRNA targets**. MiRNA targets were obtained from TargetScanMouse (targets-can.org/mmu_71) Release 7.1[31]. Targets were ranked by the total number of 8mer sites first, then total number of m8-7mer sites and finally according to the cumulative weighted context score. Only miRNAs with at least 4000 RPM in mESCs[56] were considered. Covariation enrichment was calculated for each target set using the indicated number of top-ranking targets.

**Nuclear proximity**. Nuclear gene proximity was estimated using chromosome conformation capture via high-throughput sequencing (HiC). Public preprocessed data for intrachromosomal[37] (retrieved from the HiC project website of the Bing Ren lab) and interchromosomal[38] contacts were integrated. Annotations mapping to mouse genome assembly mm9 were lifted to assembly mm10 using UCSC Genome Browser Utilities (liftOver). Normalized HiC reads for the assessment of intrachromosomal proximity were binned into 40 kilobase (kb) regions. Reads were considered supporting the proximity between two genes if their gene bodies (entire gene annotation plus a 2 kb region upstream of the gene) were each within 20 kb distance from the respective regions. Only contacts that were supported by more than one normalized read were considered. For the covariation enrichment analysis gene pairs were stratified according to their linear proximity (their absolute distance along the genome in base pairs) and according to whether there was evidence for 3D proximity based on HiC data as described above. Topologically associated domains were based on HiC experiments performed with HindIII restriction

enzyme[37]. HiC reads for the assessment of interchromosomal proximity were binned into 500 kilobase (kb) regions. Reads were considered supporting the proximity between two genes if their gene bodies (entire gene annotation plus a 2 kb region upstream of the gene) were each within 500 kb distance from the respective regions. Ranking is based on the total number of reads supporting each gene pair.

**Functional gene annotation and protein interaction**. Gene ontology and pathway annotations were retrieved from Gene Ontology project[57] (version 2.1, via MGI) and the KEGG PATHWAY Database[58] (via *KEGGREST* for R) respectively. Exceedingly small (fewer than ten genes) or big (more than 100 genes) gene ontologies were discarded. Each gene ontology and each KEGG pathway was considered as gene set for enrichment analysis. Protein interactions were retrieved from the STRING Database[59] (v10). Only interactions supported by experimental evidence were considered. For enrichment plots, gene pairs were randomly binned together (bin size 500). Further information on protein interaction in pathways and in complexes was retrieved from the Reactome project[60]. Reactome data was stratified according to whether genes interact in complex or reaction (in this paper the latter is termed pathway for reasons of consistency). Pairs were binned together randomly for enrichment plots (bin size 100).

**Regulatory and functional predictions**. Potential novel transcription factor targets are predicted as those genes that covary significantly with a certain number of the top 100 targets that were identified in one ChIP-seq experiment. Furthermore, novel predictions had to be absent from the top 1000 targets in this first experiment. Predictions success was scored as the fraction of these genes that could be identified as targets via ChIP-seq in a second experiment. For this, the top 1000 targets of this second experiment were considered. To test whether gene function can be deducted from neighboring gene pairs in a covariation network, we picked nine genes that are well characterized but act in different pathways and complexes. We then performed gene ontology enrichment with topGO on the ten genes that show the highest covariation coefficients with the gene in question. If the bona fide function of said gene was among the top ten enrichments we list the function with the associated *p*-value.

**Single-cell quantitative PCR (sc-qPCR)**. sc-qPCR was performed on preamplified cDNA (Illumina Nextera XT dual index library prep kit following the Smart-Seq2 protocol) for 21 genes and 112 cells. Applied Biosystems TaqMan assays (FAM-MGB) were used and reactions were run on a Fluidigm Biomark microfluidic qPCR chip (PN 100-6170 C1). Spearman's ranked coefficient was used for estimating covariations between sc-qPCR and single-cell RNA sequencing data. Limit of detection (LoD) was set at 39 cycles for covariations. Importantly, the results were overall consistent when using lower LoDs (incl. 25 cycles).

**Single-molecule RNA fluorescence in situ hybridization**. Custom mouse specific probes for single-molecule fluorescence in situ hybridization of gene pairs of interest were designed and produced in-house using an analytic software and production pipeline[61]. Each probe consists of 42–86 oligos (47, 51, 63, 42, and 86 oligos for the probe targeting the Adam19, Adam23, Dnmt3, Mme, and Tmem2 respectively) and each oligo consists of four parts (from 5′ to 3′): (1) a 20 nt adapter, C, for probe visualization; (2) a 20 nt adapter, F, for PCR amplification during probe synthesis; (3) a 30 nt T sequence complementary to the target; and (4) a 20 nt adapter, R, for PCR amplification during probe synthesis. The T sequences were designed using a custom-made pipeline run with the following parameters: (1) Targeting the longest transcript. (2) Having GC content between 40 and 60%. (3) Having up to 4 nt long homopolymers. (4) Allowing for at least 3 nt between consecutive oligos. The T sequences for all the probes as well as the transcript variant which they are targeting can be found in Supplementary Data 3. The C, F, and R adapter sequences were designed as explained in Gelali et al.[61] with the difference that the sequences were checked for orthogonally against the mouse genome. The probes were produced using a pipeline for large-scale enzymatic production of hundreds of probes in parallel that is describe in details in (Gelali et al.[61]). More specifically, the F and R sequences are used as barcodes for the selective amplification of the desired oligonucleotides from an array-synthesized complex oligo pool (containing thousands of different species of oligonucleotides). During the PCR reaction the C and the T7 promoter sequences are incorporated into the PCR products. The T7 promoter sequence is used for the in vitro transcription (IVT) step that follows the PCR. The RNA product of the IVT is used as a template for the reverse transcription and in the final step it is removed via alkaline hydrolysis resulting in the production of the desired ssDNA probes containing the C sequence of interest. Mouse ESCs were fixed on coverslips immobilized onto a silicon gasket. The first hybridization was performed at 37 °C for 16–18 h using a hybridization buffer containing 45% Formamide, 2× SSC, 10% Dextran sulfate, 1 mg ml⁻¹ E.coli tRNA, 0.02% bovine serum albumin, 10 mM Vanadyl-ribonucleoside complex, and the subsequent wash was performed at 37 °C using a wash buffer containing 35% Formamide, 2× SSC while the hybridization of a fluoresently labelled oligo to the C adapter was performed at 37 °C for 16–18 h using a hybridization buffer containing 30% Formamide, 2× SSC, 10% Dextran sulfate, 1 mg ml⁻¹ E. coli tRNA, 0.02% bovine serum albumin, 10 mM Vanadyl-

ribonucleoside complex, and the subsequent wash was performed at 37 °C using a wash buffer containing 25% Formamide, 2× SSC. All solutions were prepared in RNase-free water. The transcripts were probed as follows: Tmem2 with A594 and Mme with Cy5 (representing a negatively covarying gene pair); Adam19 with Cy7 (measured on the ir800 channel) and Adam23 with Cy5 (representing a positively covarying gene pair). To quantify the smFISH signal, all mRNA molecules in each field of view were counted using custom scripts in MATLAB®. To get an estimate of the mRNA counts per cell, the z-projection of the mRNA dots identified in each stack was split into a regular grid of squared pseudo-cells.

**Computation**. All computational analyses were performed in R Studio (v1.2.1335, R v3.6.1) All values (e.g., *p*-values) that are estimated to be smaller than the machine epsilon (machine precision) of $2.220446 \times 10^{-16}$ are conservatively rounded up to $2.3 \times 10^{-16}$.

**Reporting summary**. Further information on research design is available in the Nature Research Reporting Summary linked to this article.

## Data availability
Sequencing data have been deposited at NCBI SRA under the BioProject ID PRJNA592852. Gene Ontology data can be retrieved via MGI http://www.informatics.jax.org/faq/GO_dload.shtml. KEGG data was retrieved using KEGGREST in R. STRING DB data can be downloaded via http://version10.string-db.org/download/protein.links.detailed.v10/10090.protein.links.detailed.v10.txt.gz. Other data is available at the indicated locations (see "Methods" section). Preprocessed data tables can be provided upon request. All data is available from the corresponding author upon reasonable request.

## Code availability
iFISH software can be found at http://ifish4u.org. R code is available upon request.

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

## Acknowledgements

We acknowledge the following funding sources: ERC Starting Grant 758397, "miRCell"; Swedish Research Council (VR) grant 2015-04611, "MapToCleave"; and funding from the Strategic Research Area (SFO) program of the Swedish Research Council through Stockholm University. The computations were performed on resources provided by SNIC through Uppsala Multidisciplinary Center for Advanced Computing Science (UPPMAX). The Smart-Seq2 data were generated by the Eukaryotic Single Cell Genomics (ESCG) and sc-qPCR was facilitated by the Single Cell Proteomics facilities at Science for Life Laboratory. We also thank CRG (the Center for Genomic Regulation) for support in the early pilot phase of the project. Gratitude to Franziska Bonath for designing Fig. 4D and for helping with figure design. Special thanks to Marie Öhman, Johan Elf, and Claes Andréasson, as well as the Friedländer, Kutter, and Pelechano labs for their comments and suggestions.

## Author contributions

J.F., S.C.-S., D.B.M., L.Z., and M.R.F. conceived the project. S.D.M. performed sequence mapping, expression quantification and quality control and M.T. performed all further computational analyses, which were supervised by M.R.F. J.F. and S.C.-S. performed cell perturbation and sorting experiments. E.G. performed and M.B. supervised smFISH experiments. I.B. performed smFISH image analysis. I.B. and C.G. performed single-cell qPCR experiments. L.Z. performed and S.O. supervised early pilot phase computational analyses. M.T. and M.R.F. wrote the manuscript, with contributions from all authors.

## Funding

## Competing interests

The authors declare no competing interests.
