## [Peer Review File · Nature Communications]

Reviewers' comments:

Reviewer #1 (Remarks to the Author):

In their manuscript entitled “Nuclear gene proximity and protein interactions shape transcript covariances in mammalian single cells”, Tarbier et al describe a single-cell transcriptomics analysis of several hundred mouse embryonic stem cells. They use these data to identify genes with correlated mRNA expression across different cells and analyse the potential regulatory mechanisms responsible for their coexpression. While this is an interesting manuscript and a valuable dataset, I’m not sure if it provides sufficient new biological insights to warrant publication in Nature Communications.

For example, the fact that intra- and inter-chromosomal proximity often leads to gene coexpression has been reported before, although to my knowledge not in a single-cell context. Single-cell transcript coexpression studies have also been published previously, and for example the analysis by Wang et al (Plos Comput Biol, 2016) also showed that single-cell coexpression events are enriched for protein-protein interactions.

In addition, I have a number of minor comments:

1) The authors argue that having a homogenous population of cells helps to identify covariation of genes that is independent of differentiation or cell cycle progression. That makes sense. However, the paper does not seem to capitalise on that. At least, the general conclusions (e.g. covariation of targets of the same transcription factors or miRNAs) are in line with what was already known from bulk-cell coexpression studies.

2) Related to this, I wonder if the authors could justify their selection of the G2/M phase as the condition of choice, since transcriptional regulation will be much reduced during mitosis, as far as I know.

3) The authors sometimes use the term covariance when they refer to Spearman correlation (which is not the same thing), but in other places seem to actually use covariances rather than correlations. Please clarify.

4) The authors state that “67,328 gene pairs were considered significantly covarying (42,938 positively and 24,390 negatively) with an estimated false positive rate lower than 3.1% after stringent covariance calling (see Online Methods, Suppl. Table 4, 5)”. In the online methods, they refer to this “covariance calling” as a simple Spearman correlation, with a cut-off of $p < 0.01$. Where does the false positive rate come from? Moreover, without correction for multiple hypothesis testing, this is not a very stringent cut-off. The authors should also state the actual correlation coefficient ranges.

5) Related to this, the authors describe a new statistical method called “Covariance enrichment score”. Perhaps I missed something, but why is this necessary? Why not just compare the distribution of

correlation coefficients? For example, to compare two groups of genes, one could just take sample 1 (e.g. Spearman correlations between gene pairs targeted by transcription factor X) and sample 2 (e.g. random set of genes) and then test if the distributions are different using e.g. a Mann-Whitney U test.

Reviewer #2 (Remarks to the Author):

In this paper, the authors carefully characterize co-expression patterns using single cell RNA-seq on cell-cycle sorted mouse ES-cells. This population of cells is expected to be highly homogenous without active regulatory switches. Therefore, most variation is probably due to common effects in expression noise. However, based on pair-wise rank correlations the authors can still recover biologically meaningful network expression networks, which are exemplified with an representative example. I find the sectioning of regulation at the transcription as opposed to regulation by mRNA degradation interesting and it is a nice idea to use intron-counts to distinguish those effects. This also shows then in the comparison of the comparison of the Drosha-KO cells. The impact of miRNAs on expression networks appears to be a strength of this study in terms of the presented experiments. Unfortunately the narrative does not focus on this observation, but rather on the effect of nuclear proximity, which might in part be due to technical artefacts (GC..). Furthermore, it is unclear whether proximal genes merely share sources of expression noise or whether the covariance is indeed functionally meaningful.

- 1.) My strongest concern here is that the assumed Null model ignores possible correlations due to technical effects such as transcript length and GC-content. Both those parameters undoubtedly have an effect on read counts. I would like to see analyses of transcript length and GC-content on the co-expression scores. If they have an impact, a correction of the for this technical co-variance should be incorporated.
- 2.) I would like to see more QC for the single cell sequencing: How much was sequenced? I would also like to see a TSNE or UMAP to get an impression how little structure those data have left.
- 3.) I am strongly concerned with the basic processing of the data. Tophat2 is quite old and development has discontinued. In our experience, it loses many reads that can be faithfully recovered by mappers, e.g. STAR. Furthermore, in RNA-seq it is uncommon to remove duplicates using the samtools rmdup function, since this will not only remove PCR-duplicates and therefore bias results (see Parekh et al. Scientific Rep. 2016). I am also wondering about the normalisation: due to dropouts and other issues RPM is not considered to be ideal. See Vieth et al Nat. Comm. 2019 for a comprehensive evaluation of single cell normalisation methods.
- 4.) I only noted in the supplementary that batches were analysed separately, and that only correlations found in at least 2 batches were considered to be significant? This approach appears fairly ad hoc to me, wouldn't it be better to integrate across batches?

5.) Comparing the numbers of co-variances between Droscha -/- and ctrl cells, the authors find more nearly double as many losses as gains of co-variance pairs. The losses are expected, because a regulatory pathway is knocked out. I am intrigued by the many gains?? Where do they come from? Do they have any biological meaning? Or are they simply a result of a lack in power to detect such co-variations reproducibly?

7.) The stronger detected predictor of co-expression is linear proximity on the chromosome even if they are not in a TAD. The simplest possible explanation for such variation would be segregating CNVs in the mESC population. Is there anything known about this?

8.) Keeping in mind that the authors here analyse biological transcriptional noise, some of the conclusion seem unwarranted: p.7. L175-183. The pos. co-variation of ribosomal proteins observed here, does not exclude the possibility that they have a negative correlation if more distinct cell states or types are considered. Similarly, I find it difficult interpret the CES as a the relative importance of TFBS, miRNAs and chromosomal proximity. First of all, it is not the binding of a single TF that makes the expression, but it is the combinatorics and the expression levels of the TFs themselves. The same is true for miRNAs. Thus the noise in the TF and miRNA - gene association will weaken correlations, while nuclear proximity can be measured without much error. Again, if different cell states or types were compared the effects of TFs and miRNAs would most likely become stronger, probably stronger than nuclear proximity.

Minor comments.

P.5 L. 109 I disagree with the statement about technical noise in Smart-seq2: It has much more technical noise than the now commonly used UMI-methods. (E.g. Ziegenhain et al Mol. Cell. 2017)

P.5 L 116 I don't see any prove for the statement that their analysis yields reliable gene expression. Maybe correlation report how well expression correlates between cells across genes within and between batches.

P7/8 I would describe the covariance score before they are used in the section "Covariances retrace known aspects .."

P8 The Notation of the equation is not explained. Also there is a typo in the denominator $\text{signCov}(g_i, g_i)$ —> $\text{signCov}(g_i, g_j)$.

P11L327ff Be consistent in the reporting of enrichment either use % more or X-fold change.

P11L349 Did the authors make sure that to only use experimental evidence gathered in STRING — I thought most the association there come from text mining? Do they also have evidence codes?

Reviewer #3 (Remarks to the Author):

This paper by Tarbier et al. presents a broad analysis of gene expression from transcriptional to post-transcriptional regulation of mRNA, focusing chiefly on coordinated functionally related events that are quantified by patterns of covariance across single cells. In contrast to other similar studies, these investigators sought to reduce “confounding extrinsic effects” that can potentially generate extraneous variables that mask precise data. Their approach was to block cell cycle progression and limit uninduced differentiation of stem cells through culturing techniques. While one might question how effective these approaches are at suppressing these confounders and whether there are other uncontrolled sources of extrinsic variance, one cannot doubt that the remaining cells were more homogenous and much improved compared to previous studies (see citations 8-9). Moreover, the application of single cell sequencing of these suppressed cells allows improved detection of covariance of gene pairs. To investigate the covariance of large gene sets the investigators developed a scoring method called “covariance enrichment score” (CES). Using CES they found that gene sets sharing common functions are more likely to covary than randomly sampled gene sets. Furthermore, they found that gene sets comprised of proteins that physically interact are more likely to covary than gene sets of non-interacting proteins. To that end, the authors proposed comparative hypotheses and found evidence supporting the protein complex hypothesis, although there may still be some room for the alternative pathway model under different contexts or treatments. Additionally, the authors find that miRNA-mediated regulation, transcription factor-mediated regulation, and chromosomal nuclear proximity all contribute to covariance. In the case of miRNA-mediated regulation they demonstrate that this covariance is dependent on mature, processed miRNAs by using Drosha knockout cells. Indeed, except for a few minor issues, the results are quite convincing and informative. The additional strength of this study is strong confirmation of the now well-established RNA operon/regulon model of post-transcriptional regulation.

One concern is that when comparing the relative contribution of the various gene expression steps to covariance (see Fig 3 & accompanying text) is that the data underlying the categorizations is from distinct methodologies. For example, the miRNA targets were categorized based off of predictions while the TF and proximity gene sets were categorized from empirical data. These miRNA predictions frequently include false targets and miss known targets thus when it is used as a classifier in the present study it may underestimate miRNA contribution to covariance. This limitation of the analysis should be noted in the accompanying text of the results/discussion. It is also worth noting that, while not addressed in the present study, sequence-specific RNA binding proteins may also contribute to covariance. Of minor note, the equation for the CES denominator on page 8 contains two “g sub i”s while one should be “g sub j”.

Reviewer #1:

In their manuscript entitled “Nuclear gene proximity and protein interactions shape transcript covariances in mammalian single cells”, Tarbier et al describe a single-cell transcriptomics analysis of several hundred mouse embryonic stem cells. They use these data to identify genes with correlated mRNA expression across different cells and analyse the potential regulatory mechanisms responsible for their coexpression. While this is an interesting manuscript and a valuable dataset, I’m not sure if it provides sufficient new biological insights to warrant publication in Nature Communications.

For example, the fact that intra- and inter-chromosomal proximity often leads to gene coexpression has been reported before, although to my knowledge not in a single-cell context. Single-cell transcript coexpression studies have also been published previously, and for example the analysis by Wang et al (Plos Comput Biol, 2016) also showed that single-cell co-expression events are enriched for protein-protein interactions.

We agree that there is some overlap in methods between our study and that of Wang et al., and for this reason we now also cite the study in our section on protein interactions. However, to study non-genetic cell-to-cell variability, we have taken great care to remove the confounding effects of the cell cycle, differentiation and even the circadian clock. In contrast, the study by Wang et al. span over several human individuals, several cancers and many distinct and potentially heterogenous cell populations, all of which may be perturbed by genetic variation and alterations and also changes in epigenetic states caused by cancer.

Importantly, it is not clear in the Wang et al. study that the covariances that are observed in single cells are biologically meaningful. Figure 3A of that study shows that the covariances that are specific to single cells are no more enriched in shared functional GO terms than are random genes. The figure shows that covariances that are shared between single cells and bulk data are enriched in GO terms, but these shared covariances are rare, and only makes up ~7% of the top 1000 single-cell covariances (page 3 of that study). This is in contrast with our study, where our identified covarying genes are clearly enriched for biological function, with more than 200 enriched GO terms (e.g. our Figure 1F and 4C).

With regard to protein-protein interactions, we present evidence for over 3,500 gene pairs that covary at the single cell level and where the protein products physically interact, as compared to 90 gene pairs for Wang et al. This has enabled us to uncover that interaction affinity scales with covariance enrichment, to stratify them according to the type of interaction, and to quantitatively compare their impact with that of shared function (see our main Figure 4).

In addition, I have a number of minor comments:

- 1) The authors argue that having a homogenous population of cells helps to identify covariation of genes that is independent of differentiation or cell cycle progression. That makes sense. However, the paper does not seem to capitalise on that. At least, the general conclusions (e.g.

covariation of targets of the same transcription factors or miRNAs) are in line with what was already known from bulk-cell co-expression studies.

While it is true that individual regulatory factors, such as pluripotency related transcription factors, have been linked to covariance among their targets, it is surprising and novel to see this effect for hundreds of transcription factors, many miRNA and genes in nuclear proximity at the same time in the same system, without an underlying dynamic process that is observed over extended periods of time. Thus, we can for instance observe co-expression of targets of transcription factors that are linked to housekeeping processes such as transcription elongation. In classical co-expression studies such subtle effects would be masked by few strong acting and dynamic transcription factors related to the major driver of the study, e.g. differentiation or genetic heterogeneity.

2) Related to this, I wonder if the authors could justify their selection of the G2/M phase as the condition of choice, since transcriptional regulation will be much reduced during mitosis, as far as I know.

We have decided to study single mouse embryonic stem cells in the G2M phase in 2i+LIF medium, because these cells are known to be particularly transcriptionally homogenous (Kolodziejczyk et al., 2015). Furthermore, G2 and M phase are so similar in gene expression that computational tools cannot separate them, while G1 and S phase can be identified (Scialdone et al., 2015). This likely stems from the inherent delay of RNA metabolism. This delay has been estimated for mouse chromaffin cells (a type of neuroendocrine cells) to be around ~3h (Le Manno et al., 2018). It is therefore possible that the effects of transcriptional shut-down at the onset of M-phase cannot be observed during G2M-phase, but become visual in G1 phase only.

3) The authors sometimes use the term covariance when they refer to Spearman correlation (which is not the same thing), but in other places seem to actually use covariances rather than correlations. Please clarify.

We apologize for the inconsistent use of terminology. All analyses were performed on normalized covariances, and we changed the phrasing in some places to stress that we, mathematically speaking, investigated correlations, not covariances. However, when referring to the biological interpretation of the data we mainly use the term covariance, simply referring to the fact that genes show similar expression patterns across cells. In these cases, we specifically avoid the terms co-expression and correlation as they have been very much associated with bulk studies.

4) The authors state that “67,328 gene pairs were considered significantly covarying (42,938 positively and 24,390 negatively) with an estimated false positive rate lower than 3.1% after stringent covariance calling (see Online Methods, Suppl. Table 4, 5)”. In the online methods, they refer to this “covariance calling” as a simple Spearman correlation, with a cut-off of $p < 0.01$. Where does the false positive rate come from? Moreover, without correction for multiple hypothesis testing, this is not a very stringent cut-off. The authors should also state the actual correlation coefficient ranges.

The estimated false positive rate results from one million permutations of the gene expression matrix. The percentage 3.1% (corresponding to ~2000 false positives out of the 67,328 significantly correlated genes) is the highest number of significant correlations compared to the observed number in any of these permutations. The robustness of our approach stems from the use of replicates. While the false positive rate of a single correlation experiment is potentially high, the likelihood of falsely calling a significant correlation in two separate sequencing experiments is very low, as shown by our permutation test. This reduction in false positives due to the use of replicates is also the reason why abstained from using multiple hypothesis testing.

5) Related to this, the authors describe a new statistical method called “Covariance enrichment score”. Perhaps I missed something, but why is this necessary? Why not just compare the distribution of correlation coefficients? For example, to compare two groups of genes, one could just take sample 1 (e.g. Spearman correlations between gene pairs targeted by transcription factor X) and sample 2 (e.g. random set of genes) and then test if the distributions are different using e.g. a Mann-Whitney U test.

The main benefit of our covariance enrichment score (CES) is that it is a readily interpretable measurement. For instance, it is difficult to grasp the magnitude of a change in correlation coefficient e.g. by 0.2 with a p-value of 0.04. A covariance enrichment of 1.5 on the other hand can easily be understood as observing 1.5 times more significant covariances than expected by chance. This makes the results much more accessible to the reader. At the same time, it assigns a single meaningful value and thus enables the visualization of hundreds of enrichments in e.g. box- or violin plots. For illustration we provide a qq-plot of transcription factor target correlations (exemplified by Myc) compared with correlations within a random gene set as suggested by the reviewers [Tarbier_qqplot_TF]. We can clearly see transcription factor targets having higher correlation coefficients than random gene sets. We argue that while it is easy to interpret the CES, 2.22 times as many significant correlations than expected by chance, it is difficult to interpret the p-value of $< 1.4 \cdot 10^{-14}$.

qq-plot: transcription factors

Tarbier_qqplot_TF: Quantile-quantile plot of Spearman's rank coefficient for pair-wise correlations of the top 100 Myc targets compared vs. 100 randomly sampled genes. Transcription factor targets exhibit higher pair-wise expression correlation than random gene sets. P-value reflects a Mann-Whitney U test.

Reviewer #2:

In this paper, the authors carefully characterize co-expression patterns using single cell RNA-seq on cell-cycle sorted mouse ES-cells. This population of cells is expected to be highly homogenous without active regulatory switches. Therefore, most variation is probably due to common effects in expression noise.

However, based on pairwise rank correlations the authors can still recover biologically meaningful network expression networks, which are exemplified with an representative example. I find the sectioning of regulation at the transcription as opposed to regulation by mRNA degradation interesting and it is a nice idea to use intron-counts to distinguish those effects. This also shows then in the comparison to the Drosha-KO cells. The impact of miRNAs on expression networks appears to be a strength of this study in terms of the presented experiments. Unfortunately the narrative does not focus on this observation, but rather on the effect of nuclear proximity, which might in part be due to technical artefacts (GC..). Furthermore, it is unclear whether proximal genes merely share sources of expression noise or whether the covariance is indeed functionally meaningful.

What thank the reviewer for the comments and we agree that our new insights on miRNAs naturally inducing covariances transcriptome-wide are novel and well supported by the data. We have decided not to focus on the miRNA findings, since we want our study to reach a wide audience. As described below, we provide compelling evidence that our covariances are not the results of biases and are indeed functionally meaningful.

1.) My strongest concern here is that the assumed Null model ignores possible correlations due to technical effects such as transcript length and GC-content. Both those parameters undoubtedly have an effect on read counts. I would like to see analyses of transcript length and GC-content on the co-expression scores. If they have an impact, a correction for this technical co-variance should be incorporated.

We assessed potential biases due to GC content and length of genes using a binning approach. Genes were binned according to their GC content or length respectively. We then performed covariance enrichment between bins and found that genes of similar GC content and length are indeed slightly more likely to covary [Tarbier_length_and_GC_biases, column 1].

Tarbier length and GC biases: Heat maps of co-expression enrichment scores ($\log_2(\text{CES})$) in 10 bins of genes sorted according to GC content or length respectively. Genes with similar GC content or length show enrichment for co-expression. Alternative method 1: Mapping with STAR, no de-duplication, scran normalization using size factors. Alternative method 2: Same as method 1 but with additional length normalization. Alternative method 3: Same as method 2 but with hard correction of GC content and length biases in CES background model.

To address these biases, we first applied the STAR mapping, de-duplication (no de-duplication) and normalization (scran) strategy proposed by the reviewer (see Reviewer 2, comment 3 below), to all of the cells. In effect this is a complete reanalysis of all of our sequence data from the bottom-up. This did however not remove the biases, and neither did the addition of a simple length normalization regression approach [Tarbier_length_and_GC_biases, column 2 and 3]. Importantly however, using the exact mapping, deduplication and normalization strategy proposed by the reviewer did not change the overall conclusions of our study, as exemplified by the Spearman's rank correlation coefficients; by covariance enrichment of GO terms or KEGG terms; or by covariance enrichment of genes targeted by the same transcription factors or genes in nuclear proximity [Tarbier_coefficients_and_CES, left panel and right panel *'alt'*].

Tarbier_coefficients_and_CES, Left

panel: Heat scatter plot of Spearman's rank coefficients for pair-wise correlations of 200 randomly sampled genes using two different analysis pipelines. Pipeline one uses TOPHAT for mapping, applies de-duplication and an RPM normalization. Pipeline two uses STAR for mapping, applies no de-duplication and uses a length normalization in combination with scran's estimation of size factors. There is good agreement between the correlation coefficients. **Right panel:** Co-expression enrichment scores (CES) for top 50,000 gene-pairs in nuclear proximity (purple, NP) top 50 transcription factor targets (blue, TF) and all GO (orange) and KEGG pathway (yellow) annotation. First violin in each group shows original analysis (TOPHAT, de-dupl., RPM), second violin shows suggested alternative (**alt**) analysis (STAR, no de-dupl., length normalization and scran estimation of size factors), third violin shows additional hard correction/adjustment (**adj**) of GC and length biases in the CES background model. Overall, co-expression enrichments were retained a similar magnitude independent of exact analysis pipeline and GC/length bias correction.

Lastly, to ensure that we completely eliminate the effect of GC content and transcript length on the gene expression covariances, we adjusted our background CES model to correct for the biases that we observe [Tarbier_length_and_GC_biases, column 4]. For this, instead of assuming random distribution of significant correlations according to each gene's total number of significant observations, we add a factor considering its GC content and transcript length as well as the GC content and transcript length of its potential co-expression partners.

This approach combined with the exact mapping, deduplication and normalization strategy proposed by the reviewer indeed eliminates any biases, while at the same time not changing our overall conclusions [Tarbier_coefficients_and_CES, '**adj**']. We do however observe a slight drop in covariance enrichment for genes that share the same KEGG annotations, suggesting that some biological signal may have been removed.

The fact that a hard correction for GC content and transcript length seems to remove some biological signal may reflect the biological rather than technical origin of the biases. While GC

content and transcript length are known to impact transcriptomics data due to technical aspects such as amplification efficiency, they are also known to affect the transcriptome in biological meaningful ways. For instance, long transcripts are more likely to be regulated than are short house-keeping transcripts, and recently genes with low GC content have been shown to be more likely to be inefficiently translated and subsequently degraded (Courel et al., 2019). In the same study, it is shown that genes with low GC content are more likely to have regulatory functions and are more likely to be targets of miRNAs. Genes with high GC content on the other hand are preferentially decayed via NMD (Imamachi et al., 2017). We performed enrichment analysis for Gene Ontology, KEGG pathways and WikiPathways and found that genes with similar GC content or similar length are often enriched for biological functions. For instance, long genes are more likely to have functions in signal transduction while short genes often have house-keeping functions such as mRNA processing and translation. Another study (Sharova et al., 2009) estimated half-lives of mRNAs in mouse embryonic stem cell and found that long transcripts and transcripts with a low GC-content have shorter half-lives. We are therefore concerned that removing all length and GC biases may remove biological signals, while possibly introducing technical overfitting biases instead. For this reason, we prefer to keep our original analysis, while presenting the analysis with the hard correction mainly in the supplementary section (new Suppl. Figures 13 and 14).

2.) I would like to see more QC for the single cell sequencing: How much was sequenced? I would also like to see a TSNE or UMAP to get an impression how little structure those data have left.

PCA, tSNE and UMAP assign our cells to a single cluster, supporting the notion of great homogeneity within our population [Tarbier_dimensionality_reduction]. Of course, our Drosha KO cells -- which are void of miRNA -- cluster distinctly separate from the parental population, indicating that there are no cell swaps. The same results have been observed independent of the exact mapping, deduplication, and normalization strategy (not shown here).

Figure continues on next page

We further provide a table of mapping and gene expression abundance, variation and **Tarbier_dimensionality_reduction**: Dimensionality reduction of the cells considered in this study. Left column: Only parental cells, colors indicate the three sequencing runs. Right column: Parental cells (black) and Drosha KO cells (red). First row: PCA components 1 and 2. UMAP components 1 and 2.

correlation statistics across all three replicates [Tarbier_statistics].

Second row: PCA components 3 and 4. Third row: tSNE components 1 and 2. Fourth row:

m

Tarbier statistics:

Mapping statistics across the three replicates

Measurement	Replicate	Min.	25%	Median	75%	Max.
Total mapped reads	1	136638	275756	349539	430778	551584
	2	129862	225867	269602	313955	417829
	3	130947	296490	374160	425664	587444
Percentage of ERCC spike-in reads	1	0.45	0.66	0.72	0.78	1.16
	2	0.35	0.67	0.74	0.81	1.17
	3	0.38	0.65	0.72	0.78	1.10

Gene expression statistics across the three replicates

Measurement	Replicate	Min.	25%	Median	75%	Max.
Gene expr. (abundance)	1	0	0.05	2.13	15.86	1836.74
	2	0	0.04	1.59	11.89	1603.87
	3	0	0.06	2.18	16.12	1977.29
Gene expr. variation	1	0	0.38	3.51	11.60	362.28
	2	0	0.28	2.72	8.79	311.28
	3	0	0.39	3.72	12.03	452.17
Gene expr. correlation	1	-0.52	-0.02	0.06	0.17	1.00
	2	-0.58	-0.03	0.03	0.11	1.00
	3	-0.50	-0.02	0.05	0.14	1.00

Gene expression statistics across the three replicates, alternative method*

Measurement	Replicate	Min.	25%	Median	75%	Max.
Gene expr. (abundance)	1	0	0.08	2.26	15.55	4450.03
	2	0	0.05	1.68	11.85	3568.07
	3	0	0.08	2.28	15.70	5297.45
Gene expr. variation	1	0	0.47	3.70	11.90	1566.50
	2	0	0.35	2.93	9.13	1174.35
	3	0	0.49	3.93	12.31	2169.58
Gene expr. correlation	1	-0.54	-0.02	0.06	0.16	1.00
	2	-0.68	-0.03	0.03	0.12	1.00
	3	-0.67	-0.02	0.05	0.14	1.00

*Mapped with STAR, no Deduplication, normalization with SCRAN size factors

3.) I am strongly concerned with the basic processing of the data. Tophat2 is quite old and development has discontinued. In our experience, it loses many reads that can be faithfully recovered by mappers, e.g. STAR. Furthermore, in RNA-seq it is uncommon to remove duplicates using the samtools rmdup function, since this will not only remove PCR-duplicates and therefore bias results (see Parekh et al. Scientific Rep. 2016). I am also wondering about the normalisation: due to dropouts and other issues RPM is not considered to be ideal. See Vieth et al Nat. Comm. 2019 for a comprehensive evaluation of single cell normalisation methods.

As suggested we reanalyzed our data using STAR for mapping and without applying deduplication (see reviewer 2, comment 1 above). While the absolute abundance of some genes change, the overall distribution across all cells remains remarkably similar. Spearman's ranked correlation between the expression values used in this study and those retrieved by reanalysis are above 0.95 for over 95% of all genes included [Tarbier_mapping_deduplication]. Also, the overall correlation between gene pairs did not change based on the exact processing of the sequencing data. Next, we performed deconvolution normalization (estimation of size factors) for comparison with our RPM normalization, and observed that again correlations between gene pairs were highly similar. Importantly, the covariance enrichments for genes that are in nuclear proximity, that share the same transcription factors, GO term or KEGG terms are not substantially changed [Tarbier_coefficients_and_CES]. We therefore conclude that our signal is robust and not dependent on the exact mapping tool, handling of duplicates or normalization applied.

Tarbier_mapping_deduplication: Spearman's rank correlation coefficients for expression values retrieved different analysis pipelines.

4.) I only noted in the supplementary that batches were analysed separately, and that only correlations found in at least 2 batches were considered to be significant? This approach appears fairly ad hoc to me, wouldn't it be better to integrate across batches?

Looking at basic parameters of the different batches [Tarbier_statistics] we noticed that they deviate slightly from each other. Since such batch effects are likely to introduce false positive covariances we decided to analyze batches separately comparable to replicates. Requiring gene pairs to be significant in two out of three replicates greatly decreases the false positive rate, since it is unlikely for a random correlation to appear in two independent technical replicates. This approach leads to a low estimated false positive rate of 3.1% (see Reviewer 1, comment 4 above). Finally, we avoided classical batch correction/integration methods since we did not want to risk removing part of the biological signal alongside the technical biases.

5.) Comparing the numbers of co-variances between Droscha -/- and ctrl cells, the authors find more nearly double as many losses as gains of covariance pairs. The losses are expected, because a regulatory pathway is knocked out. I am intrigued by the many gains? Where do they come from? Do they have any biological meaning? Or are they simply a result of a lack in power to detect such co-variations reproducibly?

We find that genes that gain covariances in the Droscha KO are enriched in metabolic functions, in particular related to mitochondrial functions (see Supplementary Figure 9D). We do not have a clear explanation for this - possibly it has to do with cellular stress, or possibly the number of mitochondria in each cell varies more in the absence of miRNAs, causing more covariances between mitochondrial genes.

7.) The stronger detected predictor of co-expression is linear proximity on the chromosome even if they are not in a TAD. The simplest possible explanation for such variation would be segregating CNVs in the mESC population. Is there anything known about this?

The same pattern of intra-chromosomal covariances are also found in Droscha KO cells, which we clonally expanded just a few weeks before sequencing. Since these cells are almost certainly genetically identical, we can exclude that the covariances are caused by CNV variations. An alternative explanation for the relative importance of linear proximity rather than being located in the same TAD might be the epigenetic properties of mESCs. These cells are generally hypomethylated and show little variation in chromatin marks. Therefore, TADs may have a lower impact in mESCs than what would be expected from fully differentiated cells.

8.) Keeping in mind that the authors here analyse biological transcriptional noise, some of the conclusions seem unwarranted: p.7. L175-183. The pos. co-variation of ribosomal proteins observed here, does not exclude the possibility that they have a negative correlation if more distinct cell states or types are considered. Similarly, I find it difficult to interpret the CES as the relative importance of TFBS, miRNAs and chromosomal proximity. First of all, it is not the binding of a single TF that makes the expression, but it is the combinatorics and the expression

levels of the TFs themselves. The same is true for miRNAs. Thus the noise in the TF and miRNA - gene association will weaken correlations, while nuclear proximity can be measured without much error. Again, if different cell states or types were compared the effects of TFs and miRNAs would most likely become stronger, probably stronger than nuclear proximity.

We agree with the reviewer on these caveats. We originally included the statement about the ribosomal proteins as a proof-of-principle of what types of analyses can be performed within our framework, but we agree that our specific conclusions for ribosomal proteins cannot be generalized. We have rephrased the statements about the ribosomal proteins to accommodate these concerns:

‘It was recently reported that four of these proteins (RPL10, RPL38, RPS7, RPS25) are optional components of the ribosome, whose inclusion or exclusion can influence which pools of transcripts are preferentially translated ²⁰. We find that these four ribosomal proteins all covary positively and significantly with each other, providing evidence that they may not function by a mutually exclusive “either-or” logic in single mouse embryonic stem cells in steady state condition.’

Further, we have introduced a sentence in the Results and an entire Discussion paragraph discussing the caveats of comparing the relative importance of TFBS, miRNAs and chromosomal proximity (Reviewer 3, comment 1).

Minor comments.

P.5 L. 109 I disagree with the statement about technical noise in Smart-seq2: It has much more technical noise than the now commonly used UMI-methods. (E.g. Ziegenhain et al Mol. Cell. 2017)

We agree that the lack of UMIs is a weakness of Smart-seq2. However, in the same study Smart-seq2 is highlighted as the protocol with the highest sensitivity and accuracy. We rephrased the sentence accordingly.

P.5 L 116 I don’t see any proof for the statement that their analysis yields reliable gene expression. Maybe correlations report how well expression correlates between cells across genes within and between batches.

We have changed ‘reliable’ to ‘robust’. This seems reasonable given that we limit our analysis to highly expressed genes that display substantial biological variation above the technical noise as measured by the spike-ins.

P7/8 I would describe the covariance score before they are used in the section “Covariances retrace known aspects ..”

We prefer to keep the introduction to covariance enrichment score where it is, since we only investigate pair-wise covariances before this point, also in the “Covariances retrace known

aspects of stem cell biology”. Changing from studying covariances between pairs of genes, and then to sets of genes, and then back to pairs of genes again could confuse the reader.

P8 The Notation of the equation is not explained. Also there is a typo in the denominator $\text{signCov}(g_i, g_i) \rightarrow \text{signCov}(g_i, g_j)$.

We thank the reviewer for pointing this out. We have corrected this equation.

P11L327ff Be consistent in the reporting of enrichment either use % more or X-fold change.

We completely agree, and we now consistently report fold-changes, in some cases complemented also with the percentage.

P11L349 Did the authors make sure that to only use experimental evidence gathered in STRING – I thought most the associations there come from text mining? Do they also have evidence codes?

Yes, STRING also provides interactions based on experimental evidence and only those were considered. Evidence includes among others co-IP, cosedimentation and tandem affinity purification.

Reviewer #3:

This paper by Tarbier et al. presents a broad analysis of gene expression from transcriptional to post-transcriptional regulation of mRNA, focusing chiefly on coordinated functionally related events that are quantified by patterns of covariance across single cells. In contrast to other similar studies, these investigators sought to reduce “confounding extrinsic effects” that can potentially generate extraneous variables that mask precise data. Their approach was to block cell cycle progression and limit uninduced differentiation of stem cells through culturing techniques. While one might question how effective these approaches are at suppressing these confounders and whether there are other uncontrolled sources of extrinsic variance, one cannot doubt that the remaining cells were more homogenous and much improved compared to previous studies (see citations 8-9). Moreover, the application of single cell sequencing of these suppressed cells allows improved detection of covariance of gene pairs. To investigate the covariance of large gene sets the investigators developed a scoring method called “covariance enrichment score” (CES). Using CES they found that gene sets sharing common functions are more likely to covary than randomly sampled gene sets. Furthermore, they found that gene sets composed of proteins that physically interact are more likely to covary than gene sets of non-interacting proteins. To that end, the authors proposed comparative hypotheses and found evidence supporting the protein complex hypothesis, although there may still be some room for the alternative pathway model under different contexts or treatments. Additionally, the authors find that miRNA-mediated regulation, transcription factor-mediated regulation, and chromosomal nuclear proximity all contribute to covariance. In the case of miRNA-mediated regulation they demonstrate that this covariance is dependent on mature, processed miRNAs by using Droscha knockout cells.

Indeed, except for a few minor issues, the results are quite convincing and informative. The additional strength of this study is strong confirmation of the now well-established RNA operon/regulon model of post-transcriptional regulation.

We thank the reviewer for his kind comments and observations.

One concern is that when comparing the relative contribution of the various gene expression steps to covariance (see Fig 3 & accompanying text) is that the data underlying the categorizations is from distinct methodologies. For example, the miRNA targets were categorized based on predictions while the TF and proximity gene sets were categorized from empirical data. These miRNA predictions frequently include false targets and miss known targets thus when it is used as a classifier in the present study it may underestimate miRNA contribution to covariance. This limitation of the analysis should be noted in the accompanying text of the results/discussion.

We agree that a truly fair comparison of regulatory layers is not possible due to the varying ways that e.g. TF and miRNA targets were identified. We have included this sentence at the end of the relevant Results paragraph:

‘While the exact enrichments are likely specific to the study system and target selection strategy, it is clear that transcriptional regulation, miRNA-mediated regulation, and surprisingly, inter-chromosomal nuclear proximity all play important roles, and that, in our setting, intra-chromosomal nuclear proximity is the strongest predictor of gene expression covariances.’

We have further included a new Discussion paragraph that specifically highlights these caveats.

‘A caveat of our study is that different published data sources and methods were used to identify targets of miRNAs, transcription factors and genes that are in close proximity, complicating direct comparisons between them. For instance, public ChIP-seq was used to infer transcription factor targets, while Hi-C was used to detect genes in the same nuclear vicinity. These methods have distinct limitations and ranges of sensitivity. However, all of the methods that we leverage here are state-of-the-art in their respective fields, and in some cases the methods have limitations even within those areas. For instance, there is evidence that Hi-C may underestimate inter-chromosomal contacts (PubMed 29526697) and it is well-established that even the best miRNA target prediction has imperfect accuracy (PubMed 18668037 and 18668040), meaning that the effects of inter-chromosomal nuclear proximity and miRNA repression on gene expression covariances may be more profound than what we estimate here.’

With regard to miRNA target identification, we completely agree with the caveats that the reviewer points out. Computational target predictions are known to include numerous false positives and negatives, and experimental methods like CLIP-seq can include genuine interaction sites that however do not confer repression (Agarwal et al., Elife 2015). In our study we have decided to use TargetScan predictions since they have repeatedly been shown to be good predictors of miRNA repression (e.g. Baek et al., Nature 2008). We here include CDF plots that show de-repression of TargetScan miRNA targets in our Drosha KO cells [Tarbier_miRNA_CDF, next page], as a sanity check that the targets that we consider are mostly genuine targets.

It is also worth noting that, while not addressed in the present study, sequence-specific RNA binding proteins may also contribute to covariance.

We profiled two RBPs, Ptbp1 and Lin28a, and indeed their targets are substantially enriched (2.46-fold and 2.31-fold respectively) for co-expression. We find this intriguing but beyond the scope of the current study – however we may pursue this further in ongoing research.

Of minor note, the equation for the CES denominator on page 8 contains two “g sub i”s while one should be “g sub j”.

We thank the reviewer for pointing this out. We have now corrected the equation.

miRNA targets are de-repressed in a Drosha KO

Tarbier_miRNA_CDF: Cumulative distribution function of expression fold-changes between parental cells and Drosha KO cells (void of miRNA) for the top 300 targets of the most abundant miRNA in mESC and genes that are not predicted to be targets of any of these miRNAs.

REVIEWER COMMENTS

Reviewer #1 (Remarks to the Author):

I have now had a chance to read the revised manuscript of Tarbier et al entitled “Nuclear gene proximity and protein interactions shape transcript covariances in mammalian single cells”. In general, the authors have addressed my concerns in a satisfactory manner.

One remaining concern I have is the use of the term “covariance”. I understand the wish of the authors to avoid the terms “correlation” and “co-expression” to distinguish themselves from bulk studies. However, the reason these terms are used in most studies is because they happen to be the correct description of what has been measured, which holds true whether one talks about correlation in bulk or in single-cell samples. Covariance, on the other hand, is a clearly defined statistical term that refers to another type of analysis. As a compromise, I would suggest that the authors could use “covariation” when referring to biological co-expression / correlation, because that term to my knowledge is more open to interpretation than covariance.

In addition, while I found the author’s explanation for the purpose of the covariance (covariation) enrichment score convincing, I couldn’t find such an explanation in the actual manuscript text. I think it would be sensible to flesh this out to the reader as well, not just in the response to the reviewers.

Finally, I think the method to estimate the false discovery rate is questionable. Unfortunately, the online methods appear to be missing from the revised manuscript, but from the explanation in the response to the reviewers I am not sure if this is a robust approach. The methods summary states that “Pairs of genes were considered significantly covarying if they were correlated by Spearman’s ranked test ($p < 0.01$) in at least two out of three replicates.”. This is a sensible definition and I think this should be sufficient. In my opinion it’s better to drop the FDR estimation than to give a somewhat intransparent and possibly false sense of the true FDR. However, one definitely needs to apply multiple hypothesis correction to the p-values from the Spearman test when looking at this scale of interactions (for example, one could use BH correction here to define the FDR that way). If that means that fewer pairs cross the “correlated” threshold, I would say that’s ok, since this is after all a single cell study.

Reviewer #2 (Remarks to the Author):

The authors have addressed my technical concerns and indeed seem to find evidence for RNA-operons/regulons in mouse ESCs.

I still have my problems to see nuclear positioning as a means of functional regulation as stated in the abstract. ^[SEP]My interpretation is that nuclear positioning puts physical limits on the possibility to regulate one gene completely independent of its neighbouring genes and thus if tight control is necessary, genes that are also functionally related (e.g. as found in KEGG or STRING) will eventually end up in the

same 'operon'. My argument is based on first principles and difficult to test with the data, but the spirit should at least be discussed.

One way to start evaluating this would be to look at evolutionary conservation: the proximal genes that are in the same pathway should more conserved than proximal genes that just also co-expressed.

Reviewer #3 (Remarks to the Author):

I am satisfied with the authors' rebuttal to my concerns and believe the revised manuscript should be accepted. Jack D. Keene

We thank the reviewers for their comments, which we think have now resulted in a well-polished manuscript. Please find our specific responses below.

Reviewer #1 (Remarks to the Author):

I have now had a chance to read the revised manuscript of Tarbier et al entitled “Nuclear gene proximity and protein interactions shape transcript covariances in mammalian single cells”. In general, the authors have addressed my concerns in a satisfactory manner.

One remaining concern I have is the use of the term “covariance”. I understand the wish of the authors to avoid the terms “correlation” and “co-expression” to distinguish themselves from bulk studies. However, the reason these terms are used in most studies is because they happen to be the correct description of what has been measured, which holds true whether one talks about correlation in bulk or in single-cell samples. Covariance, on the other hand, is a clearly defined statistical term that refers to another type of analysis. As a compromise, I would suggest that the authors could use “covariation” when referring to biological co-expression / correlation, because that term to my knowledge is more open to interpretation than covariance.

The suggestion to use covariation to avoid confusion and any inaccuracy is very much appreciated. We now apply mainly the term “covariation” throughout the manuscript, and occasionally the term “co-expression”, thus completely avoiding the term “covariance”.

In addition, while I found the author’s explanation for the purpose of the covariance (covariation) enrichment score convincing, I couldn’t find such an explanation in the actual manuscript text. I think it would be sensible to flesh this out to the reader as well, not just in the response to the reviewers.

We agree that a motivation for the introduction of this metric is beneficial for the readers and therefore added this sentence in the manuscript.

“The CES provides an easily interpretable single metric – fold-enrichment rather than difficult to interpret coefficients and p-values – that allows for easy visualization and comparison of covariations.”

Finally, I think the method to estimate the false discovery rate is questionable. Unfortunately, the online methods appear to be missing from the revised manuscript, but from the explanation in the response to the reviewers I am not sure if this is a robust approach. The methods summary states that “Pairs of genes were considered significantly covarying if they were correlated by Spearman’s ranked test ($p < 0.01$) in at least two out of three replicates.”. This is a sensible definition and I think this should be sufficient. In my opinion it’s better to drop the FDR estimation than to give a somewhat intransparent and possibly false sense of the true FDR. However, one definitely needs to apply multiple hypothesis correction to the p-values from the Spearman test when looking at this scale of interactions (for example, one could use BH correction here to define the FDR that way). If that means that fewer pairs cross the “correlated” threshold, I would say that’s ok, since this is after all a single cell study.

We agree that using Spearman’s ranked test and applying correction for multiple hypothesis testing to the resulting p-values is the most straightforward way to estimate significance. For this purpose, we pooled our three replicate plates of cells, calculated Spearman’s p-values and used Benjamini-Hochberg as suggested for correction, resulting in 252,146 significant gene pairs (see table below). In comparison, our approach is more ad hoc, being tailored for our specific experimental setup that uses 3 replicate plates of cells. We found that the 81,833 gene pairs that result from our method are almost all (89.8%) included in the set of 252,146 gene pairs. In other words, our method can be seen as a more stringent approach, but importantly almost all of the gene pairs that we report are also reported by the approach suggested by the reviewer. The same holds when we apply scran batch correction to the replicates first (see table below).

We could in principle apply the proposed approach to our study, but this would in effect mean that every data point, every figure and every table would need to be redone. Given that our gene pairs are essentially all contained in the set produced by the approach suggested by reviewer, we suggest that we keep the current approach, but add these sentences to the Results section:

“As an additional benchmark, we performed correlation calculation on the pooled replicates and applied multiple-hypothesis testing (Benjamini-Hochberg). Around 90% of the covariations called by our approach are supported by pooled and corrected covariations. In fact, our approach is stricter than simple multiple-hypothesis testing since fewer gene pairs are considered significant.”

We agree with the reviewer that the term false discovery rate can be misleading when applied to permutation tests. We have therefore rephrased the sentence in the Results section:

“We randomly permuted the count matrix one thousand times and found that the highest number of significant covariations observed was ~2000, only 3.1% of the covariations observed in the original data (see Online Methods, Suppl. Table 5, 6).”

preprocessing	pooled uncorrected (p<0.05)	pooled corrected (p<0.05, BH*)	this study [‡]	overlap	overlap in % of this study
this study	1,407,435	252,146	81,833	73,466	89.8%
batch corr.	1,414,653	288,571	107,661	98,693	91.7%

* Benjamini-Hochberg; [‡] p<0.01 in two replicates and same sign in all three replicates; gene pairs include ribosomal proteins and are therefore slightly higher than those in the manuscript

Reviewer #2 (Remarks to the Author):

The authors have addressed my technical concerns and indeed seem to find evidence for RNA-operons/regulons in mouse ESCs. I still have my problems to see nuclear positioning as a means of functional regulation as stated in the abstract. My interpretation is that nuclear positioning puts physical limits on the possibility to regulate one gene completely independent of its neighbouring genes and thus if tight control is necessary, genes that are also functionally related (e.g. as found in KEGG or STRING) will eventually end up in the same 'operon'. My argument is based on first principles and difficult to test with the data, but the spirit should at least be discussed. One way to start evaluating this would be to look at evolutionary conservation: the proximal genes that are in the same pathway should more conserved than proximal genes that just also co-expressed.

We agree that the logic of "co-regulation" of closely localized genes might be a reverse one, where genes acting in the same complex and therefore needing co-expression are evolutionary more likely to localize in close proximity. However, even when comparing genomes of model organisms, it is difficult to test this hypothesis, since there are additional factors impacting the likelihood of conserved co-localization, such as the exact distance in nucleotides, the localization on the chromosome (e.g. close to a telomere vs. close to the centromere) and the presence of repeat sequences in the proximity of the gene pair of interest. Additionally, the assignment of the closest homologs is not always biunique. We therefore agree that it is the most feasible to discuss such evolutionary aspects in the Discussion section. We there add the following:

"It is possible that nuclear proximity limits independent regulation of physically close genes rather than enabling active co-regulation. Genes that need tight co-regulation may therefore evolutionary tend to locate in proximity on the same chromosome. Independent of the causality, genes that share a protein interaction and therefore need stoichiometric co-expression may be placed in nuclear proximity through evolution."

Reviewer #3 (Remarks to the Author):

I am satisfied with the authors' rebuttal to my concerns and believe the revised manuscript should be accepted. Jack D. Keene

We are delighted to hear that all concerns have been addressed.

REVIEWERS' COMMENTS:

Reviewer #1 (Remarks to the Author):

The authors have now addressed my remaining concerns in full and I am now happy to recommend this article for publication.